# A Parameterization of Heterogeneous Hydrolysis of $N_2O_5$ for Mass-based Aerosol Models: Improvement of Particulate Nitrate Prediction

Ying Chen[1,2], Ralf Wolke[1], Liang Ran[3], Wolfram Birmili[1,4], Gerald Spindler[1], Wolfram Schröder[1], Hang Su[2,5], Yafang Cheng[2,5], Ina Tegen[1] and Alfred Wiedensohler[1]

[1]Leibniz-Institute for Tropospheric Research (TROPOS), Leipzig, 04318, Germany
[2]Multiphase Chemistry Department, Max Planck Institute for Chemistry, Mainz, 55128, Germany
[3]Key Laboratory of Middle Atmosphere and Global Environment Observation, Institute of Atmospheric Physics, Chinese Academy of Sciences, Beijing, 100029, China
[4]German Environment Agency, Dessau-Roßlau, 06844, Germany
[5]Institute for Environmental and Climate Research, Jinan University, Guangzhou, China

*Correspondence to*: Y. Chen (chen@tropos.de) and R. Wolke (wolke@tropos.de)

## Abstract

Heterogeneous hydrolysis of $N_2O_5$ on the surface of deliquescent aerosol particles leads to $HNO_3$ formation and acts as a major sink of NOx in the atmosphere during nighttime. The reaction constant of this heterogeneous hydrolysis is determined by temperature (T), relative humidity (RH), aerosol particle composition as well as the surface area concentration (S). However, these parameters were not comprehensively considered in the parameterization of heterogeneous hydrolysis of $N_2O_5$ in previous mass-based 3-D aerosol modelling studies. In this investigation, we propose a sophisticated parameterization (NewN2O5) of $N_2O_5$ heterogeneous hydrolysis with respect to T, RH, aerosol particle compositions and S, based on laboratory experiments. We evaluated closure between NewN2O5 and a state-of-the-art parameterization based on a sectional aerosol treatment. The comparison showed a good linear relationship (R=0.91) between these two parameterizations. NewN2O5 was incorporated into a 3-D fully online coupled model: COSMO-MUSCAT with the mass-based aerosol treatment. As a case study, we used the data from the HOPE-Melpitz campaign (10-25 September 2013) to validate model performance. Here, we investigated the improvement of nitrate prediction over the western and central Europe. The modelled particulate nitrate mass concentrations ($[NO_3^-]$) were validated by filter measurements over Germany (Neuglobsow, Schmücke, Zingst, and Melpitz). The modelled $[NO_3^-]$ were significantly overestimated for this period by a factor of 5-19, with the corrected $NH_3$ emissions (reduced by 50%) and the original parameterization of $N_2O_5$ heterogeneous hydrolysis. The NewN2O5 significantly reduces the overestimation of $[NO_3^-]$ by ~35%. Particularly, the overestimation factor was reduced to approximately 1.4 in our case study (September 12, 17-18 and 25, 2013), when $[NO_3^-]$ was dominated by local chemical formations. In our case, the suppression of organic coating was negligible over western and central Europe, with an influence on $[NO_3^-]$ less than 2% on average and 20% at the most significant moment. To obtain a significant impact of the organic coating effect, $N_2O_5$, SOA and $NH_3$ are needed to be present when RH is high and T is low. However, those conditions were rarely fulfilled simultaneously over western and central Europe. Hence, the organic coating effect on reaction probability of $N_2O_5$ may not be as significant as expected over western and central Europe.

## 1 Introduction

The budget of nitrogen oxides (NOx) is of fundamental importance for tropospheric chemistry (Ehhalt and Drummond, 1982). The most important removal path of nitrogen from the atmosphere is the formation of $HNO_3$, which is transferred to particles or deposited eventually (Riemer et al., 2003). $HNO_3$ is mainly produced via the reaction of $NO_2$ and OH at daytime. At nighttime, the heterogeneous hydrolysis of $N_2O_5$ on the surface of deliquescent aerosol particles forming $HNO_3$ is a major sink of NOx (Jacob, 2000; Brown and Stutz, 2012; Platt et al., 1984; Brown et al., 2004). Given that NOx is the key precursor of ozone, chemical mechanisms controlling the budget of NOx also have an important impact on ozone and oxidizing capacity of the atmosphere on a global scale (Dentener and Crutzen, 1993; Evans and Jacob, 2005).

The reaction constant of the hydrolysis of $N_2O_5$ ($k_{N_2O_5}$) on the surface of deliquescent aerosol particles can be quantified by the reaction probability ($\gamma_{N_2O_5}$). It has been measured for surfaces of different aqueous solutions by several techniques (Mozurkewich and Calvert, 1988; Van Doren et al., 1990; Fenter et al., 1996; Robinson et al., 1997; Behnke et al., 1997; Hu and Abbatt, 1997; Hallquist et al., 2000). They reported that typical values for $\gamma_{N_2O_5}$ are in the order of $10^{-2}$. Organic coating of the particles may reduce this reaction probability. Anttila et al. (2006) proposed a parameterization (Anttila06) that described the organic coating suppression effect on $\gamma_{N_2O_5}$. Based on laboratory experiments and Anttila06, Gaston et al. (2014) reported that the suppression of $\gamma_{N_2O_5}$ by organic coating is dependent on a range of factors including the O:C ratio, the organic particle mass fraction and the relative humidity (RH). Bertram and Thornton (2009) developed a parameterization to describe the influence of chloride salts on $\gamma_{N_2O_5}$ as a function of RH. However, the influence of temperature was not considered in their study. Recently, Morgan et al. (2015) demonstrated that this influence of chloride may not be represented properly, and the "high" O:C regime defined in Gaston et al. (2014) was scarcely observed over northwestern Europe by airborne regional measurement. Several laboratory studies reported that $\gamma_{N_2O_5}$ substantially varies with temperature, RH, and particle composition (Mentel et al., 1999; Kane et al., 2001; Hallquist et al., 2003; Chang et al., 2011; Brown and Stutz, 2012; Gaston et al., 2014). Davis et al. (2008) derived a parameterization (Davis08) of $\gamma_{N_2O_5}$ on the surface of particles containing ammonium, sulfate and nitrate. It was developed on basis of numerous previous laboratory studies (Mozurkewich and Calvert, 1988; Hu and Abbatt, 1997; Folkers, 2002; Kane et al., 2001; Folkers et al., 2003; Hallquist et al., 2003; Badger et al., 2006), with respect to temperature, RH and particle compositions.

Several studies have implemented the heterogeneous hydrolysis of $N_2O_5$ in global and regional chemical transport models, in order to investigate its influences on atmospheric chemistry. Dentener and Crutzen (1993) investigated the importance of the heterogeneous hydrolysis of $N_2O_5$ on a global scale, by assuming a constant value $\gamma_{N_2O_5} = 0.1$, which might be overestimated. Chang et al. (1987) proposed a scheme to parameterize the $k_{N_2O_5}$ for 3-D models without complex aerosol treatments. Riemer et al. (2003) reported that the scheme of Chang et al. (1987) could only be representative of heavily polluted conditions or if cloud droplets are at presence, with a particle surface area concentration of 2700 $\mu m^2/cm^3$. Riemer et al. (2003) proposed a more

complex scheme (P1 in the literature) with respect to the particle surface area concentration (S) and $\gamma_{N_2O_5}$ of nitrate and sulfate, which were reported by Mentel et al. (1999) and Wahner et al. (1998). However, the influence of temperature and RH on $\gamma_{N_2O_5}$ was still not considered in the P1 of Riemer et al. (2003). Later, by applying Anttila06 to the P1 of Riemer et al. (2003), Riemer et al. (2009) found that organic coatings could decrease particulate nitrate concentrations by up to 90% where both $N_2O_5$ and secondary organic compounds were built-up. Evans and Jacob (2005) developed a parameterization scheme (EJ05) that has an extensive description of aerosol composition to improve the GEOS-CHEM simulations. EJ05 included $\gamma_{N_2O_5}$ of dust (Bauer et al., 2004), sea salt (Sander et al., 2003), sulfate (Kane et al., 2001; Hallquist et al., 2003), elemental carbon (EC, Sander et al., 2003) and organic carbon (OC, Thornton et al., 2003), also took into account the dependence of $\gamma_{N_2O_5}$ on RH. However, $\gamma_{N_2O_5}$ of nitrate and its dependence on temperature were not carefully considered in EJ05. Archer-Nicholls et al. (2014) incorporated Bertram and Thornton (2009) into WRF-Chem. Lowe et al. (2015) further took the organic coating effect into account by applying Anttila06 to Bertram and Thornton (2009). However, $\gamma_{N_2O_5}$ with respect to EC, OC and dust was lacking in Bertram and Thornton (2009).

Furthermore, as mentioned above, the reported influence of chloride on $\gamma_{N_2O_5}$ (Bertram and Thornton, 2009) may not be realistic in northwestern Europe (Morgan et al., 2015). Recently, Chang et al. (2016) improved the P1 (Riemer et al., 2003) with 'Davis08+Anttila06' scheme, and incorporated it into WRF-Chem with a sectional aerosol treatment (MOSAIC, Zaveri et al., 2008). They validated the improved P1 with the aircraft measurements from CalNex 2010 campaign. 'Davis08+Anttila06' showed a better result than that from the scheme according to Bertram and Thornton (2009), and significantly improved the model performance (Chang et al., 2016). However, the influences of black carbon (BC), sea salt aerosol (SSA) and dust were still missing in the parameterizations according to Chang et al., (2016). The P1 scheme (Riemer et al., 2003; Chang et al., 2016) is very helpful for models with complex aerosol treatments (modal/sectional aerosol approach, e.g. WRF-Chem with MOSAIC). However, it can not be easily adopted in the computationally efficient mass-based aerosol approaches, which are commonly used in atmospheric chemistry transport or climate models, e.g. EMEP (Simpson et al., 2012) and GEOS-Chem (Walker et al., 2012), as well as long-term modelling studies (e.g. Bellouin et al. 2011; Hardiman et al., 2017). Riemer et al. (2003) also improved a simplified scheme (P2 in the literature) based on the work of Chang et al. (1987), which is easily adopted in the mass-based aerosol models and is currently used in COSMO-MUSCAT (**Co**nsortium for **S**mall-scale **Mo**delling and **Mu**lti-**S**cale **C**hemistry **A**erosol **T**ransport, http://projects.tropos.de/cosmo_muscat, Wolke et al., 2004; Wolke et al., 2012) with the mass-based aerosol treatment according to Simpson et al., 2003. However, the P2 still showed a large difference in comparison to the more-complete P1 (Riemer et al., 2003). The reasons may be the missing of complex considerations of S and $\gamma_{N_2O_5}$ in the P2.

To improve the representativeness of heterogeneous hydrolysis of $N_2O_5$ in 3-D models with mass-based aerosol treatment, we propose a new parameterization (NewN2O5) with respect to temperature, RH, particle composition and particle surface area. This NewN2O5 was validated by the state-of-the-art parameterization in Chang et al. (2016). We also incorporated NewN2O5 into the 3-D fully on-line coupled model COSMO-MUSCAT, in order to investigate the improvement of particulate nitrate prediction. The measurements of the

HOPE campaign (HD(CP)[2] Observational Prototype Experiment, Macke et al., 2016) at Melpitz site (12.93$^o$E, 51.53$^o$N, 86 m a.s.l., a regional background observatory of central Europe) and other three stations of the German federal environmental agency (UBA) over Germany were used to validate the simulation results.

## 2 Data & Methods

### 2.1 The model system COSMO-MUSCAT

The online-coupled chemical transport model COSMO-MUSCAT is qualified for process studies as well as the operation forecast of pollutants in local and regional areas (Heinold et al., 2011; Hinneburg et al., 2009; Stern et al., 2008; Renner and Wolke, 2010). Two nested domains with 50 vertical layers were used for this model study. The outer domain covers the whole Europe, with a spatial grid resolution of 14×14 km. The inner domain (N2) covers Germany, the Netherlands and near-by regions, with a spatial grid resolution of 7×7 km (Fig. 1). The simulation period was divided into overlapping short-term cycles. Each of these cycles consisted of a one-day spin-up for the meteorology followed by a two-day coupled run of meteorology and chemistry transport. The main features of the model system are described below. More details are given in Wolke et al. (2004, 2012) and Baldauf et al. (2011).

An adequate modelling of dynamics requires an online coupling between the chemical transport model MUSCAT and the meteorological model COSMO. Here, the compressible non-hydrostatic flow in a moist atmosphere is described by the primitive hydro-thermodynamical equations (Steppeler et al., 2003; Doms et al., 2011a). The vertical diffusion is parameterized by a level 2.5 closure scheme based on a prognostic equation for turbulent kinetic energy (Doms et al., 2011b). Moist convection is parameterized according to Tiedtke (1989). A two-stream formulation (Ritter and Geleyn, 1992) is applied for radiative transfer. Aerosol particles, clouds and tracers gases are considered as optically active constituents, modifying the radiative fluxes by absorption, scattering and emission. The meteorological forcing of COSMO is performed by reanalysis data of the German Weather Service DWD, which are derived from the global meteorological model GME (Majewski et al., 2002).

MUSCAT describes the transport, chemical and removal processes. The chemical mechanism RACM-MIM2 (Karl et al., 2006; Stockwell et al., 1997) with 87 species and more than 200 reactions is applied to represent the gaseous chemistry. A simplified mass based approach (similar to EMEP model, Simpson et al., 2003) is used to represent the aerosol processes with high efficiency. The formation of secondary inorganic particulate matter is performed by reactions between ammonia and sulfuric or nitric acid, which are produced from the gas phase precursors $SO_2$ and NOx (Hinneburg et al., 2009). The applied particle/gas partitioning depends on temperature and humidity. As in ISORROPIA (Nenes et al., 1998), the equilibrium is shifted towards the gas phase for dry and warm conditions. The implementation of this partitioning scheme is comparable to Galperin and Sofiev (1998) by using the equilibrium approach of Mozurkewich (1993). The extended SORGAM (Schell et al., 2001, Li et al., 2013) is coupled with the mass-based aerosol approach to predict the formation of secondary organic aerosol (SOA). Dry deposition is modelled by using the resistance approach described by Seinfeld and Pandis (2006), considering the atmospheric turbulence state, the kinetic viscosity, and the gravitational settling of particles. The aerodynamic and quasi-laminar layer resistances are taken from COSMO by analogy with the deposition of water vapour. The wet deposition is parameterized in dependence on the size resolved scavenging and collection efficiency (Simpson et al., 2003).

The European anthropogenic emission inventory and the temporal resolved emission factors are provided by TNO for the AQMEII project (Pouliot et al., 2012; Wolke et al., 2012). The inventory includes the gaseous pollutants (CO, NOx, $SO_2$, $NH_3$ methane and non-methane volatile organic compounds) and primary emitted particulate matters ($PM_{2.5}$, $PM_{2.5-10}$, organic carbon-OC and elemental carbon-EC) with a spatial resolution of $0.125^o \times 0.0625^o$ (lon-lat, about $7 \times 7$ km). Note that EC and BC are usually interchangeable in modelling studies (Vignati et al., 2010; Chen et al., 2016a; Nordmann et al., 2014). The emission of $NH_3$ was reduced by 50%, since over 90% of $NH_3$ emissions in Europe are contributed by agricultural sources (Hertel et al., 2011; Erisman et al., 2008; Reidy et al., 2008) and agriculture emissions of $NH_3$ are overestimated by ~50% or even more (Sintermann et al., 2012; Backes et al., 2016). Also, Chen et al. (2016b) adopted the same $NH_3$ emission inventory in the WRF-Chem model and reported that total $NH_3$ was overestimated by a factory of ~2 at Melpitz during the campaign period. The modelled dust emissions depend on surface wind friction velocities, surface roughness, soil particle size distribution, and soil moisture (Heinold et al., 2011). Sea salt emissions are parameterized depending on salinity and wind speed (Long et al., 2011). Biogenic emissions depends on land-use and meteorology by the approach of Steinbrecher et al. (2009) and for "soil NO" by Williams et al. (1992) and Stohl et al. (1996). Saarikoski et al. (2007) scheme was applied to estimate the biomass burning emissions.

The chemical inactive tracers (T1, T2 and T3) were added into RACM-MIM2 to investigate the chemical fluxes of the selected reactions. T1, T2 and T3 (representing R1, R2 and R3 respectively) were reset to zero every hour in the simulation in order to quantify the chemical fluxes from $N_2O_5$ to nitrate avoiding the influence of transport. T1 represents the contribution of $N_2O_5$ on chemical formation of nitric acid; T3-T2 represents local chemical net formation of particulate nitrate.

$$N_2O_5 + H_2O \overset{(aerosol)}{\rightarrow} 2HNO_3 + T1 \tag{R1}$$

$$NH_3 + HNO_3 \rightarrow NH_4NO_3 + T2 \tag{R2}$$

$$NH_4NO_3 \rightarrow NH_3 + HNO_3 + T3 \tag{R3}$$

Furthermore, in order to investigate the influence of transport from the $NH_3$ source regions (the Netherlands and southern Germany) on particulate nitrate, the marker tracer (T-NH3) was emitted into the first layer of regions with high ammonia emissions (white bars in Fig. 1) with an emission rate of $2 \times 10^{-8}$ mol/m$^2$/s.

**2.2 A novel mass-based parameterization of heterogeneous hydrolysis of $N_2O_5$**

The reaction of $N_2O_5$ with water vapour is very slow, therefore a considerable loss of $N_2O_5$ is assumed to occur on the surface of deliquescent aerosol particles (Platt et al., 1984), as shown in R1. Many laboratory experiments have shown that $\gamma_{N_2O_5}$ depends principally on particle composition and water content (and so atmospheric RH). Reducing the RH, or adding organics or nitrate to the solutions, suppresses $\gamma_{N_2O_5}$ by an order of magnitude (Chang et al., 2011 and references therein).

The P2 of Riemer et al. (2003), which adapted from Chang et al. (1987), was originally incorporated in COSMO-MUSCAT to represent the heterogeneous hydrolysis of $N_2O_5$ (OldN2O5). Reaction R1 is implemented

into chemical transport models as a first-order loss (Riemer et al., 2003). The reaction constant ($k_{N_2O_5}$) is defined as:

$$k_{N_2O_5} = \frac{1}{4} \cdot v_{N_2O_5} \cdot S \cdot \gamma_{N_2O_5} \tag{1}$$

where $v_{N_2O_5}$ is the mean molecular velocity of $N_2O_5$, and S is the surface area concentration of aerosol particles.

5    Based on the first-order reaction constant, Chang et al. (1987) proposed the following scheme to represent $k_{N_2O_5}$.

$$k_{N_2O_5} = \frac{1}{600\exp(-(\frac{RH}{28})^{2.8})+a} \tag{2}$$

RH is the relative humidity in %, which was used as an indicator for the influence of hygroscopic growth on S, $k_{N_2O_5}$ results in min$^{-1}$, and 'a=5' was originally adopted in Chang et al. (1987). According to Riemer et al.

10   (2003), 'a=5' represents the surface area concentration of 2700 μm$^2$/cm$^3$, when RH is higher than 60%. However, this high surface area concentration can only be expected in highly polluted areas or if cloud droplets are present. Therefore, we adopted 'a=17' in this study as suggested by Riemer et al. (2003).  However, the complex considerations of S and $\gamma_{N_2O_5}$ are still missing in the OldN2O5. In this study, we propose a sophisticated parameterization to improve the OldN2O5 for mass-based aerosol models, with a full consideration of

15   temperature, RH, aerosol particle compositions and S.

As shown in equation (1), $k_{N_2O_5}$ is linearly related to S and $\gamma_{N_2O_5}$. We adapted equation (2) with factors $f_s$ and $f_{\gamma_{N_2O_5}}$, which represent the impact of S and $\gamma_{N_2O_5}$ respectively, as shown in equation (3). $f_s$ can be estimated from the particle mass concentration, according to equation (4). $f_{\gamma_{N_2O_5}}$ can be estimated from the core-shell model of aerosol particles considering the secondary organic coating effect according to Anttila et al. (2006) and

20   Riemer et al. (2009), as given in equation (5). The influence of O:C ratio on the organic coating effect (Gaston et al., 2014) was not considered here, since the O:C ratio information is not available in COSMO-MUSCAT. Also the "high" O:C regime defined in Gaston et al. (2014) may not represent the northwestern Europe case compared with airborne measurements (Morgan et al., 2015).

$$k_{N_2O_5} = \frac{1}{600\exp(-(\frac{RH}{28})^{2.8})+17} \cdot f_s \cdot f_{\gamma_{N_2O_5}} \tag{3}$$

$$f_s = (SA_{fine} \cdot PM_{fine} + SA_{coarse} \cdot PM_{coarse}) / S_{ref} \tag{4}$$

where $SA_{fine}$ / $SA_{coarse}$ is the specific surface area for fine/coarse mode particles in m²/g, $PM_{fine}$ / $PM_{coarse}$ is the mass concentration of fine/coarse mode particles in μg/m³. A value 11 m²/g was used for $SA_{fine}$, considering recently reported values of 11.9 m²/g and 10.2 m²/g from laboratory studies (Okuda, 2013) and measurements in Japanese urban regions (Hatoya et al., 2016). A value of 1.2 m²/g was used for $SA_{coarse}$ (Okuda, 2013). $S_{ref}$ is the reference particle surface area concentration, here, we suggest ' $S_{ref}$ = 600 μm²/cm³'. Since equation (2) will provide a result that is very close to a complex parameterization with a particle surface area concentration of 600 μm²/cm³ (Riemer et al., 2003), when 'a=17' and 'RH>60%'. Note that a small initial overestimation of particulate nitrate may result in a significant overprediction of nitrate, through the integration in models due to a feedback in this scheme. That is higher nitrate concentrations result in a larger $f_s$ and promise a higher $k_{N_2O_5}$, leads to a higher production of nitrate. In order to avoid the uncertainty of this feedback mechanism and to calculate a reasonable $k_{N_2O_5}$ in this case study, the nitrate mass concentration in equation (4) is considered as 1.3 times of sulfate mass concentration based on filter measurements during the HOPE-Melpitz campaign. .

$$f_{\gamma_{N_2O_5}} = (\gamma_{core}^{-1} + \gamma_{coating}^{-1})^{-1} / \gamma_{ref} \tag{5}$$

where $\gamma_{core}$ is the $N_2O_5$ reaction probability with the core of the particle, which can be estimated by Table 1; $\gamma_{coating}$ is the $N_2O_5$ reaction probability with the secondary organic coating shell of the particle, which can be estimated by the formula (6) according to Anttila et al. (2006) and Riemer et al. (2009); $\gamma_{ref}$ is the reference reaction probability. Here, we suggest ' $\gamma_{ref}$ =0.1', since equation (2) is developed on basis of the assumption ' $\gamma_{N_2O_5}$ = 0.1' (Riemer et al., 2003).

$$\gamma_{coating} = \frac{4RTH_{N_2O_5} D_{N_2O_5} R_{core}}{v_{N_2O_5} l_{shell} R_{particle}} \tag{6}$$

where R is the universal gas constant, T is the temperature, $H_{N_2O_5}$ is the Henry's Law constant of $N_2O_5$ for the organic coating, and $D_{N_2O_5}$ is the diffusion coefficient of $N_2O_5$ in the organic coating, $R_{core}$ is the radius of the core, $R_{particle}$ is the radius of the particle, and $l_{shell}$ is the thickness of the organic coating shell.

$\gamma_{core}$ can be estimated from previous laboratory experiments (Table 1) of inorganic and primary organic compositions (Davis et al. 2008; Evans and Jacob, 2005, and references therein;). Davis et al. (2008) proposed an extended parameterization for $N_2O_5$ hydrolysis on ammonium-sulfate-nitrate particles, with respect to RH and temperature. Evans and Jacob (2005) provided the parameterizations for $N_2O_5$ hydrolysis on primary organic particles (Thornton et al., 2003), black carbon (Sander et al., 2003), sea salt (Sander et al., 2003) and dust (Bauer

et al., 2004). $\gamma_{core}$ can be derived by a mass-weighted average (Riemer et al., 2003) of each single-component parameterization (Table 1).

**2.3 Estimation of reaction probabilities with a sectional aerosol model**

The Weather Research and Forecasting/Chemistry model (WRF-Chem V3.5.1) is a fully on-line coupled regional air quality model. Chang et al. (2016) incorporated several parameterizations for the $N_2O_5$ hydrolysis into a sectional aerosol treatment (MOSAIC, Zaveri et al., 2008) in WRF-Chem. 'Davis' approach from Chang et al. (2016), hereinafter referred to as Ch&Davis, was chosen to be compared with NewN2O5. The reasons for this choice will be discussed in detail in section 3.1.

In order to validate the mass-based NewN2O5 with the sectional-based Ch&Davis, we performed WRF-Chem simulation during the HOPE-Melpitz campaign. The same WRF-Chem results were adopted for offline estimating $k_{N_2O_5}$ according to NewN2O5 and Ch&Davis, respectively. We followed the physics relating configuration according to Chen et al. (2016a), which well reproduced meteorological conditions during the HOPE-Melpitz campaign. The sea salt emission (Gong, 2003) was reduced by a factor of 20 in WRF-Chem, considering that Gong (2003) may highly overestimate sea salt emission (Neumann et al., 2016), and thus leads to an overestimation of sea salt by a factor of 20 during the HOPE campaign at Melpitz (Chen et al., 2016b). The configuration of chemical and aerosol treatments followed Chang et al. (2016). CBMZ (Zaveri and Peter, 1999) mechanism was used to describe gas-phase reactions. MOSAIC (Zaveri et al., 2008) with eight size bins was chosen to represent aerosol properties. Three nested domains (Fig. S1) with 39 vertical layers were set up for the simulated case, with a resolution of 54 km, 18 km and 6 km respectively.

In Ch&Davis the aerosol liquid water is considered when calculating particle surface area for each size bin. Details of the sectional-based method for estimating S in Ch&Davis scheme are given by Chang et al. (2016). In NewN2O5 scheme, the first six bins (with diameter in the range of 40nm – 2.5 μm) are counted as fine mode, and the last two bins (2.5 -10 μm) are counted as coarse mode. This definition is identical with COSMO-MUSCAT. In order to be consistent with COSMO-MUSCAT, the organic coating effect is considered for fine particles in NewN2O5, since the maximum effective particle diameter of Anttila06 scheme is 2 μm (Anttila et al., 2006). In order to quantify the uncertainty stem from the different S treatments between NewN2O5 (mass-based) and Ch&Davis (sectional-based), an estimation result according to an adapted NewN2O5 (with sectional-based S) will also be discussed in section 3.1.

**2.4 Observations**

The filter chemical composition measurements of the HOPE-Melpitz campaign (10-25 September 2013) and at three UBA stations (Neuglobsow, Schmücke, and Zingst, www.umweltbundesamt.de) were used to validate the modelled results. The observations at the TROPOS research station Melpitz represent the regional background of central Europe (Spindler et al., 2012; Spindler et al., 2010; Brüggemann and Spindler, 1999; Poulain et al., 2011; Birmili et al., 2001). During the HOPE-Melpitz campaign, high volume samplers DIGITEL DHA-80 (Walter RiemerMesstechnik, Germany), with a sampling flow of about 30 m$^3$/h, were used to collect 24-hour daily filter samples with 10 μm cutoff inlets. Additionally, 24-hour filter sampler measurements with PM$_{10}$ inlet at 3 UBA station in Germany were collected every third day. The filter material is quartz fibre (Munktell, Grycksbo,

Sweden, Type MK 360), which allows the determination of particle mass, water-soluble ions ($SO_4^{2-}$, $NO_3^-$, $NH_4^+$, $Cl^-$, $Na^+$, $K^+$, $Mg^{2+}$ and $Ca^{2+}$), OC and EC from one filter. The filters were pre-heated before sampling for at least 24 hours at 105 ℃ to minimize the blank values of OC. More details about filter measurement are given in (Spindler et al., 2013). Near-ground meteorological parameters (e.g. temperature, relative humidity, wind speed, wind direction) were simultaneously measured at Melpitz. More details about the HOPE campaign are given in Macke et al. (2016).

## 3. Results & Discussion

The COSMO-MUSCAT model performance was examined by comparing simulated meteorological fields with the Melpitz near-ground measurements (Fig. 2). Generally, the meteorological conditions during the HOPE-Melpitz campaign were well captured by the model, with correlation coefficients (R) of 0.87, 0.85, 0.73, and 0.85 for temperature, RH, 10-meter wind speed and wind direction, respectively. The factors between modelled results and the meteorological measurements were ~1, except for an overestimation of wind speed with a factor of 1.44, possibly due to the vertical resolution of the model. Nevertheless, the temperature and RH, which are the most important meteorological parameters in this study for $N_2O_5$ heterogeneous hydrolysis during nighttime, were in a good agreement with the measurement. Although model simulations slightly underestimated RH during the nighttime of September 17 and 22 (Fig. 2b), modelled RH was still higher than 80% where $k_{N_2O_5}$ is insensitive to RH as shown in Table 1 and Riemer et al. (2003). Therefore, this bias of RH will not lead to a significant uncertainty in nitrate simulation. However, the overestimation of wind speed may favour the transport of ammonia from Western Europe (e.g. the Netherlands). This could be a possible reason for the nitrate overprediction in NewN2O5 case (Fig. 3d), especially during September 20-24 when western wind was constantly dominant (Fig. 2d).

### 3.1 Evaluating closure for mass-based NewN2O5 and a sectional approach

In order to confirm that the mass-based NewN2O5 estimates $k_{N_2O_5}$ with a reliable accuracy, we evaluated closure between NewN2O5 and a sectional-based state-of-the-art parameterization (Chang et al., 2016) based on the WRF-Chem (MOSAIC) results. Chang et al. (2016) reported that 'Davis + coat' (Daivs08 + Anttila06) approach produced a best agreement of $\gamma_{N_2O_5}$ with aircraft observations during the CalNex-2010 campaign, with overestimation by a factor mostly within in a range of 2-8 (Fig. S2b). Without considering OC coating effect (Davis08 only), the Ch&Davis still showed a relatively good linear relationship with the observed $\gamma_{N_2O_5}$, which was however overestimated with a higher factor ranging about 3-10 (Fig. S2a). Considering the different treatments of OC coating between NewN2O5 (SOA coating only) and Chang et al. (2016) ('Davis + coat', Primary OC and SOA), the NewN2O5 was validated using the Ch&Davis scheme. This would not significantly influence the comparison results, since the HOPE-Melpitz campaign was an OC-low case, with only ~7% contribution from total OC mass based on filter measurements at Melpitz. Therefore, not much SOA was available for coating effect, different to the OC-high case (contributed about 50-80% to total mass, Figure 9 in Chang et al., 2016) in the CalNex-2010 campaign. The coating effect exerted a negligible influence at Melpitz, this point will be discussed in detail in section 3.4. We validated NewN2O5 scheme by comparing $k_{N_2O_5}$ instead

of $\gamma_{N_2O_5}$, because NewN2O5 scheme was developed on basis of a parameterization to directly calculate $k_{N_2O_5}$ proposed by Chang et al. (1987) and Riemer et al (2003).

As shown in Fig. 4, the $k_{N_2O_5}$ showed a good linear relationship (R=0.91) between NewN2O5 and Ch&Davis, much better than using the OldN2O5 (Fig. 4). Mass-based NewN2O5 estimated lower $k_{N_2O_5}$ than the sectional-based Ch&Davis by a factor of ~8. However, Ch&Davis may overestimate the $\gamma_{N_2O_5}$ by a factor of 3-10 (Chang et al, 2016, see also Fig. S2a). Assuming that S was correctly given by the WRF-Chem sectional aerosol module, we can expect that Ch&Davis may overestimate $k_{N_2O_5}$ by a factor of 3-10 according to the equation (1). Therefore, NewN2O5 may provide a $k_{N_2O_5}$ in the range of 0.36-1.2 times of the realistic one.

Two important uncertainties are needed to be kept in mind in this validation. First, the estimation of S is very challenging, due to the uncertainties of particle number/mass size distribution, partitioning processes, secondary formation and etc. In addition, the hygroscopic grow of particle can also be an important source of the uncertainty of S, due to the challenge in the estimation of particle liquid water especially at low RH, even by a complex aerosol treatment (Chang et al., 2016). About 30% difference of $k_{N_2O_5}$ between NewN2O5 and Ch&Davis is stem from the different treatments of S. As shown in Fig. 4, the factor between NewN2O5 and Ch&Davis reduced from ~8.3 to ~5.9, with a slightly increase of R, when we adopted the sectional-based S (same as Ch&Davis) in NewN2O5. Second, the Ch&Davis was validated by aircraft measurements in an OC-high case during the CalNex-2010 campaign. Therefore, the overestimation factor of Ch&Davis may not be as high as expected in an OC-low case during the HOPE-Melpitz campaign. However, the SSA, BC and dust should exert a sensible influence in an OC-low case, and should also be considered in a parameterization, as we did in NewN2O5. This can be also a reason for the difference between Ch&Davis and NewN2O5.

### 3.2 Improvement of the particulate nitrate prediction

In previous evaluation studies (Im et al., 2015; Wolke et al., 2012), the COSMO-MUSCAT model predicted particulate nitrate mass concentrations ([$NO_3^-$]) in a fair agreement with the measurements, with an overestimation in the range of 50% on long-term average. This is comparable with other models (Im et al., 2015). However, short periods with strong overestimations of [$NO_3^-$] were also observed in these previous studies. This seems to be the case for the HOPE-Melpitz campaign simulation, where COSMO-MUSCAT highly overpredicted [$NO_3^-$] over Germany in this study (Fig. 3) as well as WRF-Chem in a previous study (Chen et al, 2016b). In order to evaluate the improvement of NewN2O5 scheme and quantify the influence of $NH_3$ emission overestimation on the particulate nitrate prediction, three sensitivity simulations were conducted (Table 2).

In this HOPE-Melpitz campaign case, the particulate nitrate mass concentrations were overestimated by factors of 23.7, 12, 6.5 and 6.3 for Neuglobsow, Schmücke, Zingst, and Melpitz, respectively (Fig. 3). The modelled NOx was in line with the observed concentration level at Melpitz, and should not be the reason of the overprediction of particulate nitrate (see details in Supplement Text S1 and Fig. S3). Nevertheless, the

overestimation of $NH_3$ emission might contribute about 20-30% of the particulate nitrate overprediction, compared between OldN2O5-FullNH3 and OldN2O5 cases. This is in line with the previous studies (Renner and Wolke, 2010; Backes et al., 2016). However, even with a 50% reduction of $NH_3$ emissions, the particulate nitrate was still highly overestimated over Germany with factors of about 19, 9, 4.5 and 5 for these four stations, respectively. The NewN2O5 scheme would further moderate the overprediction by another ~35% (Fig. 3). Correspondingly, the overestimation factors of particulate nitrate were reduced to about 10.7, 6, 2.5 and 3 for the four stations, respectively. The $N_2O_5$ was almost all consumed by the heterogeneous reaction at Melpitz in OldN2O5 case, but not in the NewN2O5 case (Fig. 3e). It is due to a significant decrease (by averagely more than a factor of 20, see Fig. 4) in the reaction constant of heterogeneous hydrolysis of $N_2O_5$ by NewN2O5. However, there must be other reasons that might explain the remained overestimations in the simulated particulate nitrate mass concentrations. One possible reason can be the underprediction of coating organic matter budget in the model leading to an overestimation of $\gamma_{N_2O_5}$ (Chang et al., 2016); other possible reasons should be investigated in future studies, e.g. deposition process, long-range transport, formation of nitrogen-containing OC and neutralization processes.

The improvement of particulate nitrate prediction with NewN2O5 can be more clearly shown associated with the tracers (T1 in Fig. 3f; T3-T2 in Fig. 3g, and T-NH3 in Fig. 3h) and the comparison with Melpitz measurements (Fig. 3d), which were sampled on the filter every day and analysed off-line. The overestimation of [$NO_3^-$] in September 10-11 (grey shaded period in Fig. 3) stemmed from the uncertainty of boundary conditions in the model. As shown in Fig. S4, an air mass with high [$NO_3^-$] was transported from the southwestern boundary area to Melpitz. The [$NO_3^-$] at Melpitz was dominated by the transport from the Netherlands and southern Germany on September 13-14 and 19-24 (blue shaded period in Fig. 3), as indicated by the high T-NH3 concentration (Fig. 3h) and the negligible local chemical formations (Fig. 3g). In contrast, the local chemical formations dominated the [$NO_3^-$] in September 12, 17-18 and 25 (red shaded period in Fig. 3). During the red shaded period, T-NH3 was almost zero (Fig. 3h) and the modelled wind speed was less than 4 m/s in average (Fig. 2c). A much stronger reduction on the overestimation of particulate nitrate occurred during the red shaded period (a factor of ~1.4 in average), which was dominated by the local chemical formations. This further confirmed the improvement of heterogeneous hydrolysis of $N_2O_5$ by NewN2O5. During September 15-16 (without shaded period in Fig. 3), the contributions from both transport and local chemical formations of particulate nitrate were very limited (Fig. 3f-h), resulting in a very low [$NO_3^-$].

**3.3 Comparison between NewN2O5 and OldN2O5**

The NewN2O5 case improved the particulate nitrate overestimation problem compared with OldN2O5. Meanwhile, the spatial distribution pattern of [$NO_3^-$] was similar between these two cases (Fig. 5). Here, we focus on the nighttime period of the HOPE-Melpitz campaign, since the $N_2O_5$ heterogeneous reaction is much more significant during the night than in the daytime. The lowest [$NO_3^-$] was found over Poland and ocean regions during nighttime, [$NO_3^-$] was lower than 4 $\mu g/m^3$ and 3 $\mu g/m^3$ in OldN2O5 and NewN2O5 cases, respectively. Moderate [$NO_3^-$] was found over central Europe (Melpitz and the surrounding region), about 6-8 $\mu g/m^3$ and 4-5.5 $\mu g/m^3$ in the OldN2O5 and NewN2O5 cases, respectively. The highest [$NO_3^-$] occurs over the region of the Netherlands and near-by regions, about 9-12 $\mu g/m^3$ and 6-8 $\mu g/m^3$ in OldN2O5 and NewN2O5 cases, respectively, due to the high agriculture emission of $NH_3$ in this region. There was also a remarkably high

amount of particulate nitrate over southern Germany, about 8-10 μg/m$^3$ and 5-6.5 μg/m$^3$ in the OldN2O5 and NewN2O5 cases, respectively. In general, the [NO$_3^-$] was reduced by ~35% over the entire N2 domain (Fig. 5). The most significant reduction of [NO$_3^-$] is found over the Netherlands and southern Germany where the highest [NO$_3^-$] (reduced by about 3-4.5 μg/m$^3$) was found, followed by the near Melpitz region (reduced by about 2-3 μg/m$^3$, Fig. 5c). This is caused by a significant reduction (by more than a factor of 20, see Fig. 4) of $k_{N_2O_5}$, which is resulted from the consideration of particle mass concentration's influence on S and comprehensive treatments for $\gamma_{N_2O_5}$. Therefore, the regions with high [NO$_3^-$] during nighttime indicates a considerable nitrate formation from the heterogeneous hydrolysis of N$_2$O$_5$, where [NO$_3^-$] was reduced by about 3-4.5 μg/m$^3$ (~35%, see Fig. 5) in the new scheme. However, this heterogeneous hydrolysis was negligible over the regions where [NO$_3^-$] was low during nighttime, and did not have relevant contribution on the formation of particulate nitrate. Hence, the improvement of particulate nitrate prediction by NewN2O5 was more significant over the high-[NO$_3^-$] regions than the low-[NO$_3^-$] regions.

### 3.4 Influence of organic coating on the N$_2$O$_5$ heterogeneous hydrolysis

The secondary organic coating on particle surface may significantly decrease the reaction probability of N$_2$O$_5$ and influence the particulate nitrate concentration. Riemer et al., (2009) reported that organic coating could decrease [NO$_3^-$] by up to 90% where both N$_2$O$_5$ and secondary organic compounds were built-up. The highest reduction over Europe was found over the Netherlands followed by western Germany (both covered by the domain N2) in their study. In addition to N$_2$O$_5$ and secondary organic compounds, the meteorological conditions (e.g. RH and temperature) may also exert a sensible influence on organic coating effect. In this study, we introduced a parameterization (NewN2O5) for heterogeneous hydrolysis of N$_2$O$_5$ considering meteorological conditions. The influence of the organic coating suppression effect on particulate nitrate prediction was investigated by a comparison between NewN2O5 with and without SOA coating effect.

At nighttime, much higher N$_2$O$_5$ concentrations occured and the heterogeneous hydrolysis is more important than that during daytime (Jacob, 2000). As shown in Fig. 6a and Fig. 6b, the influence of the organic coating effect was negligible over the domain N2 including the Netherlands and Germany. Even at 24 September 23:00 CET when changes were most significant, the organic coating reduced [NO$_3^-$] only by about 2-4 μg/m$^3$ (less than 10-20%) over the black-polygon marked and near-by regions (Fig. S5). Meanwhile, for nighttime averages during the campaign, the organic coating could only reduce [NO$_3^-$] by less than 0.1 μg/m$^3$ or 2% over the whole domain (Fig. 6). This is because appropriate meteorological conditions, as described following, are needed in NewN2O5 for a significant organic coating. In addition to the simultaneous build-up of SOA and N$_2$O$_5$ (Riemer et al., 2009), high NH$_3$ concentrations and $\gamma_{N_2O_5}$ are also indispensable conditions for a significant organic coating effect. High NH$_3$ concentrations are necessary for neutralizing the HNO$_3$, which came from the heterogeneous hydrolysis of N$_2$O$_5$ during the night. High $\gamma_{N_2O_5}$ causes a significant reduction of $\gamma_{N_2O_5}$ by organic coating (Chang et al., 2016; Riemer et al., 2009). Therefore, a large impact should be expected in the regions with high RH and low temperature, hence a high $\gamma_{N_2O_5}$. As show in Fig. 6, the most significant organic coating effect (still less than 2% influence on [NO$_3^-$]) could be found over the Netherlands and near-by regions (black polygon). Over this area, these five conditions were fulfilled to some extent: (1) temperature was 13.5-14.5 $^o$C; (2) RH was

higher than 75%; (3) SOA concentration was ~1.6 μg/m$^3$; (4) $N_2O_5$ concentration was about 0.4-0.6 μg/m$^3$; (5) $NH_3$ concentration was about 4-6 μg/m$^3$ (Fig. 1). There was almost no influence of organic coating over the other regions (Fig. 6a and Fig. 6b). These five conditions (not very high temperature; relatively high RH; built-up of SOA, $N_2O_5$ and $NH_3$) could not be simultaneously fulfilled over the western and central Europe, therefore the organic coating effect was not very significant.

## 4 Conclusions

Generally, the COSMO-MUSCAT model predicted particulate nitrate in a reasonable range in long-term average. The results were comparable with other models in previous studies. However during the HOPE-Melpitz campaign (10-25 September 2013), particulate nitrate was significantly overestimated by the COSMO-MUSCAT model over Germany, despite a good performance of meteorological simulation. This can be partly (~35%) attributed to the parameterization of heterogeneous hydrolysis of $N_2O_5$ (OldN2O5). A sophisticated mass-based parameterization of heterogeneous hydrolysis of $N_2O_5$ (NewN2O5) was proposed in this study, aiming at improving the particulate nitrate prediction in atmospheric modelling. This mass-based NewN2O5 was validated with a state-of-the-art parameterization (Chang et al., 2016), which is based on a sectional aerosol treatment. The validation results showed a good linear relationship (R=0.91) and indicated that NewN2O5 could estimate the reaction probability of $N_2O_5$ in a reasonable range, within about 0.36-1.2 times of the realistic one.

In order to quantify the improvement of the nitrate prediction by using NewN2O5, sensitivity studies were performed based on more realistic $NH_3$ emissions, which are reduced by 50%. This correction was based on previous studies that showed $NH_3$ emissions were overestimated by a factor of ~2. The overestimation of $NH_3$ emissions led to about 20-30% overprediction of particulate nitrate over Germany. The horizontal distribution patterns of particulate nitrate were in a good agreement between OldN2O5 and NewN2O5 cases. OldN2O5 case overestimated particulate nitrate by a factor of 19, 9, 4.5 and 5 for Neuglobsow, Schmücke, Zingst, and Melpitz, respectively. This may be caused by lacking of consideration of particle surface area (S) and complex treatments of $\gamma_{N_2O_5}$. Based on many previous laboratory experiments, the influences of temperature, RH, aerosol particle compositions and surface area concentration on the heterogeneous reaction constant of $N_2O_5$ were considered in NewN2O5. The reaction constant was averagely reduced by a factor of more than 20 in NewN2O5. Correspondingly, the overestimation of particulate nitrate was reduced by ~35% for the whole period. Particularly, the NewN2O5 significantly improved particulate nitrate prediction, with a factor of ~1.4 compared with the filter measurements, when particulate nitrate was dominated by the local chemical formations at Melpitz (September 12, 17-18 and 25).

In this study, we additionally investigated how the decrease of $\gamma_{N_2O_5}$ due to organic coating (Anttila et al., 2006) influences the particulate nitrate prediction over western and central Europe. Based on NewN2O5, the simulation results with and without organic coating were analyzed. Our results showed a negligible (less than 2% or 0.1 μg/m$^3$) influence of coating on particulate nitrate over the Netherlands and Germany. Since, in addition to the considerable amounts of $N_2O_5$, SOA and $NH_3$ must be present at the same location, appropriate meteorological conditions (relatively high RH and low temperature) are also indispensable for the organic coating to exert a sensible impact. This is because low RH and high temperature would lead to a low $\gamma_{N_2O_5}$ value, and thereby no

significant organic coating suppression on $\gamma_{N_2O_5}$ would be observed. These conditions were rarely fulfilled simultaneously over western and central Europe; hence, the influence of the organic coating effect on particulate nitrate prediction was negligible in this study.

This study suggests that temperature, RH, particle compositions and surface area concentration should be taken into account in the parameterization of the heterogeneous hydrolysis of $N_2O_5$. A sophisticated parameterization is proposed for the mass-based aerosol models. It should be included in model simulations to improve the representativeness of the $N_2O_5$ hydrolysis of in the ambient atmosphere. The results also implicate that the organic coating effect on suppressing the heterogeneous hydrolysis of $N_2O_5$ may not be as significant as expected over Europe.

**Acknowledgements:** The HOPE campaign was funded by the German Research Ministry under the project number 01LK1212 C. We would like to thanks TNO (the Netherlands) and the AQMEII project (http://aqmeii.jrc.ec.europa.eu/aqmeii2.htm) provide the European anthropogenic emission inventory, and German federal environmental agency (UBA) provide the filter measurements of particle compositions. Furthermore, the JSC Jülich and the DWD Offenbach supported the work by providing computing time and meteorological data. We also would like to thanks Prof. Dr. Ulrich Pöschl, Prof. Dr. Hartmut Herrmann and Dr. Andreas Tilgner for the discussion and help.

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

**Table 1.** Representation of reaction probability of aerosol particle core ($\gamma_{core}$) for N$_2$O$_5$ hydrolysis.

| Particle Type | Parameterization | Parameters | References and Remarks |
|---|---|---|---|
| **Core of particle** | $\gamma_{core} = \sum_i \gamma_i \cdot ratio_i$ | $ratio_i = \dfrac{[m_i]}{[m_{core}]}$ <br><br> $[m_{core}] = \sum_i [m_i]$ <br><br> i: the following particle types <br><br> i = [ASN, OC, SSA, Dust, BC] | $\gamma$ : reaction probability <br><br> m: mass <br><br> [mass]: mass concentration |
| ASN: <br> (A) Ammounium <br> (S) Sulfate <br> (N) Nitrate | $\gamma_{ASN} = \gamma^*_{AB} \cdot x_{AB} + \gamma^*_{aq/d,AS} \cdot x_{AS} + \gamma^*_{AN} \cdot x_{AN}$ <br><br> $\gamma^*_{AB} = \min(\gamma_{AB}, 0.08585)$, $\quad \gamma_{AB} = \dfrac{1}{1+e^{-\lambda_{AB}}}$ <br><br> $\gamma^*_{aq,AS} = \min(\gamma_{aq,AS}, 0.053)$, $\gamma_{aq,AS} = \dfrac{1}{1+e^{-\lambda_{aq,AS}}}$ <br><br> $\gamma^*_{d,AS} = \min(\gamma_{d,AS}, 0.0124)$, $\gamma_{d,AS} = \dfrac{1}{1+e^{-\lambda_{d,AS}}}$ <br><br> $\gamma^*_{AN} = \min(\gamma_{AN}, 0.0154)$, $\quad \gamma_{AN} = \dfrac{1}{1+e^{-\lambda_{AN}}}$ <br><br> $\lambda_{AB} = \beta_{10} + \beta_{11} \cdot RH + \beta_{12} \cdot T_{291}$ <br><br> $\lambda_{aq,AS} = (\beta_{10} + \beta_{20}) + \beta_{11} \cdot RH + (\beta_{12} + \beta_{22}) \cdot T_{291}$ <br><br> $\lambda_{d,AS} = \beta_{d0} + \beta_{d1} \cdot RH + \beta_{d2} \cdot T_{293}$ <br><br> $\lambda_{d,AN} = \beta_{30} + \beta_{31} \cdot RH$ | $x_{AB} = 1 - (x_{AS} + x_{AN})$ <br><br> $x_{AS} = \max(0, \min(1 - x_{AN}, \frac{[A]}{[N]+[S]} - 1))$ <br><br> $x_{AN} = \dfrac{[N]}{[N]+[S]}$ <br><br> $\beta_{10} = -4.10612$ <br> $\beta_{11} = 0.02386$ <br> $\beta_{12} = -0.23771$ <br> $\beta_{20} = -0.80570$ <br> $\beta_{22} = 0.10225$ <br> $\beta_{30} = -8.10774$ <br> $\beta_{31} = 0.04902$ <br> $\beta_{d0} = -6.13376$ <br> $\beta_{d1} = 0.03592$ <br> $\beta_{d2} = -0.19688$ <br> $T_{291} = \max(T - 291, 0)$ <br> $T_{293} = \max(T - 293, 0)$ | Davis et al. (2008) <br><br> *AB: ammonium bisulfate* <br> *AS: ammonium sulfate* <br> *AN: ammonium nitrate* <br> *A: NH$_4^+$* <br> *S: SO$_4^{2-}$* <br> *N: NO$_3^-$* <br> *Unit of RH: %* <br> *Unit of T: K* <br> *aq: aqueous phase* <br> *d: dry phase (crystallized)* <br> *AS crystallizes when RH<32.8% and forms a sold phase (Martine et al., 2003)* |
| Organic Carbon (Primary OC) | $\gamma_{OC} = RH \times 5.2 \times 10^{-4} \quad RH < 57\%$ <br><br> $\gamma_{OC} = 0.03 \qquad\qquad\quad RH \geq 57\%$ | | *Evans and Jacob (2005)* <br><br> *Thornton et al. (2003)* |
| Sea Salt Aerosol (SSA) | $\gamma_{SSA} = 0.005 \qquad RH < 62\%$ <br><br> $\gamma_{SSA} = 0.03 \qquad\; RH \geq 62\%$ | | *Evans and Jacob (2005)* <br><br> *Sander et al. (2003)* |
| Dust | $\gamma_{Dust} = 0.01$ | | *Evans and Jacob (2005)* <br><br> *Bauer et al. (2004)* |
| Black Carbon (BC) | $\gamma_{BC} = 0.005$ | | *Sander et al. (2003)* |

**Table 2.** Sensitivity simulation cases

| Case | N$_2$O$_5$ parameterization | NH$_3$ emission |
|---|---|---|
| OldN2O5-FullNH3 | P2 of Riemer et al. (2003) | 100% |
| OldN2O5 | P2 of Riemer et al. (2003) | 50% [*] |
| NewN2O5 | New scheme (this study) | 50% [*] |

[*] Suggested by Sintermann et al. (2012), Backes et al. (2016) and Chen et al. (2016b)

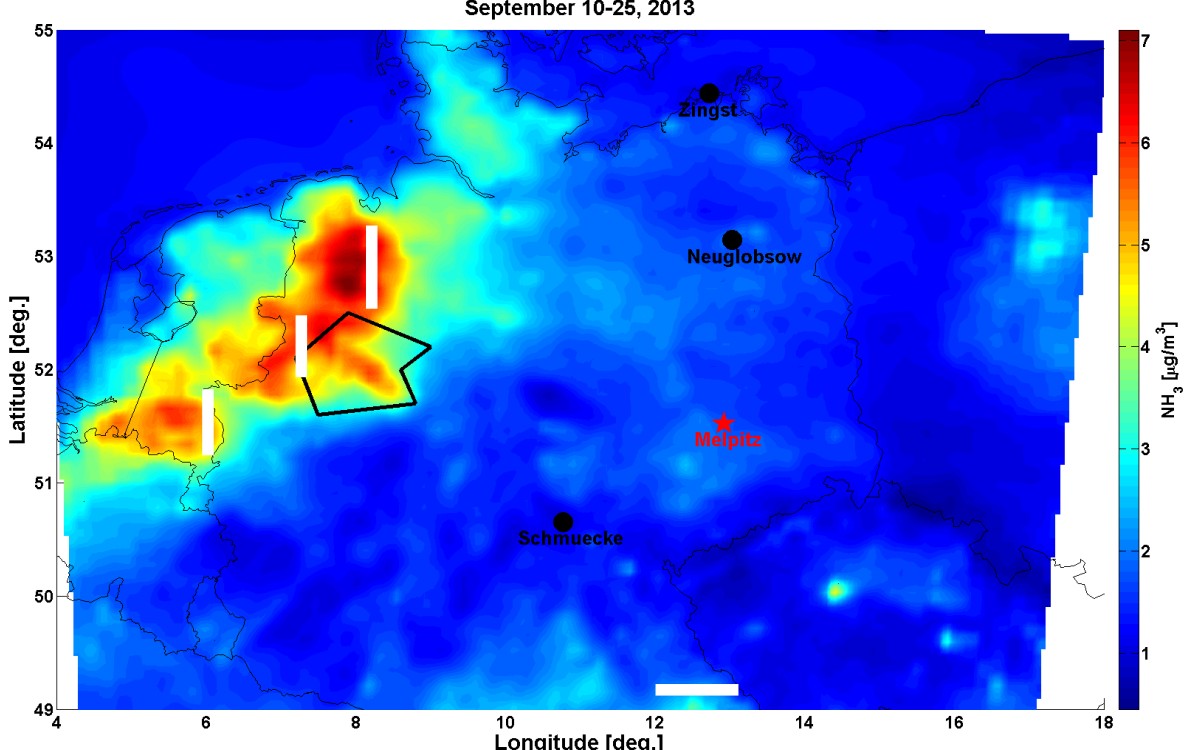

**Figure 1.** Results (domain N2) of NewN2O5 case of averaged NH$_3$ mass concentration during 10-25 September 2013. The added emissions of marker tracer (T-NH3) from NH$_3$ source regions (the Netherlands and south Germany) are marked by the white bars. The locations of the considered measurement stations are also marked: Neuglobsow, Schmücke and Zingst are marked by black dots; Melpitz is marked in a red star and its results will be detailed discussed in Fig. 2 and Fig. 3. The region with the most significant organic coating effect is highlighted by the black polygon, and will be analysed together with Fig. 6.

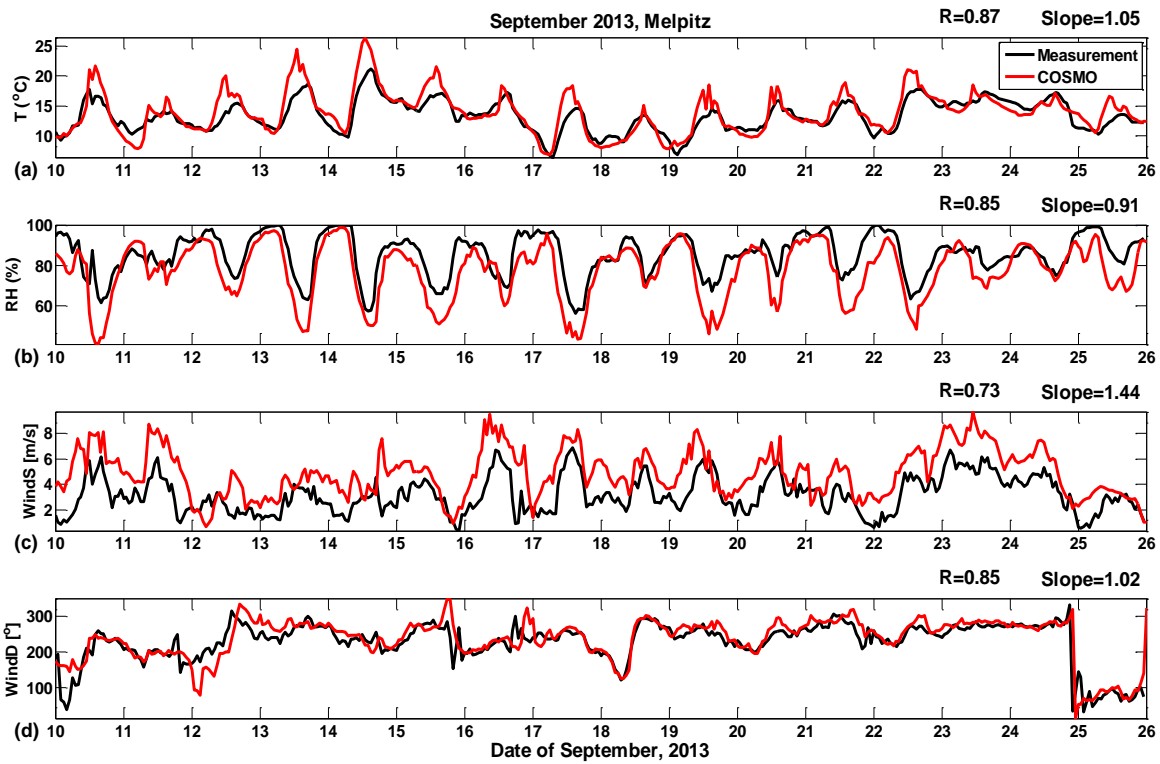

**Figure 2.** Comparison between modelled and measured meteorological conditions. (a) Temperature (T); (b) relative humidity (RH); (c) wind speed; (d) wind direction.

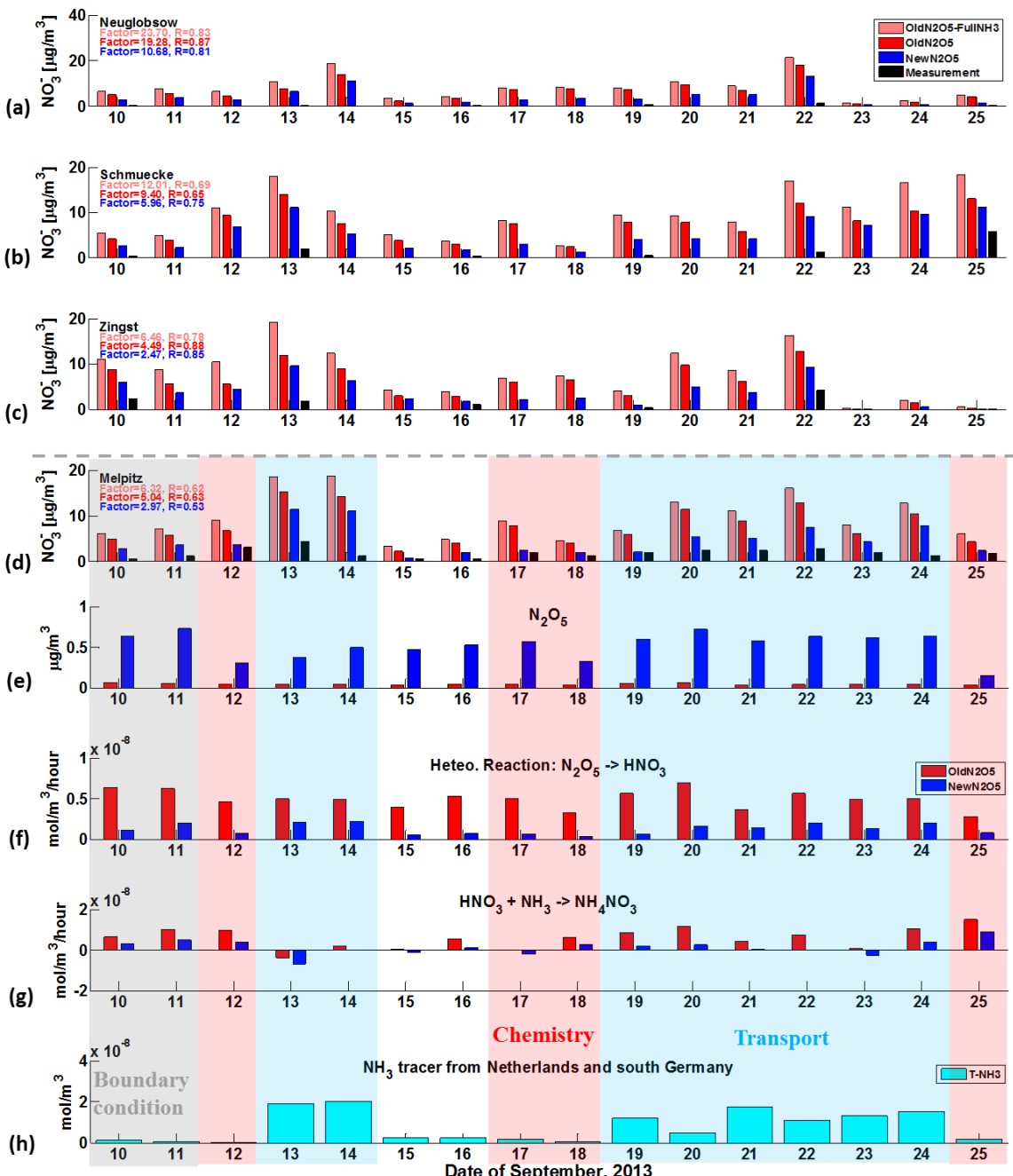

**Figure 3.** Comparison of particulate nitrate mass concentration between filter measurements and modelled results: (a) Neuglobsow; (b) Schmuecke; (c) Zingst; (d) Melpitz. Modelled concentrations at Melpitz: (e) N₂O₅; (f) marker species T1 for chemical reaction R1; (g) marker species for chemical formation of particulate nitrate (T3-T2); (h) the NH₃ marker tracer (T-NH3) for transport from the Netherlands and south Germany. The light-red colour bars indicate the results of OldN2O5-FullNH3 case; the red colour bars indicate the results of OldN2O5 case; and the blue colour bars indicate the results of NewN2O5 case. The shaded periods indicate the dominating processes for high concentrations of particulate nitrate: chemical formation (red), transport (blue), and boundary conditions (grey).

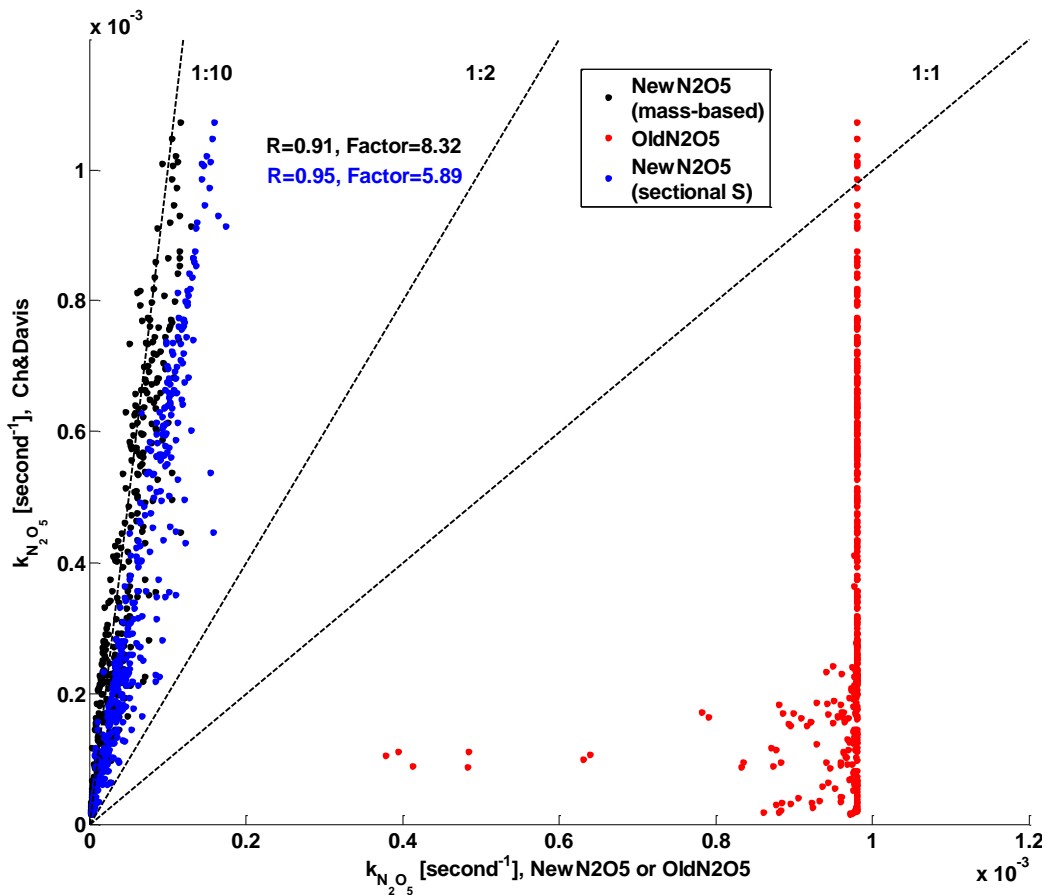

**Figure 4.** Comparison between the sectional-based Ch&Davis ('Davis' of Chang et al., 2016) and mass-based NewN2O5 (mass-based, black), NewN2O5 (with a sectional-based particle surface area, blue) and OldN2O5 (red). The results are offline calculated on basis of WRF-Chem simulation with a sectional aerosol treatment (MOSAIC).

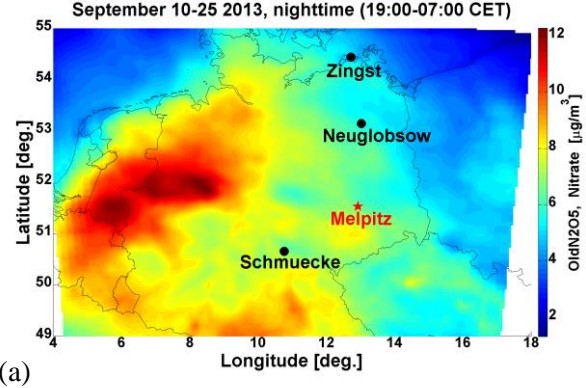

(a)

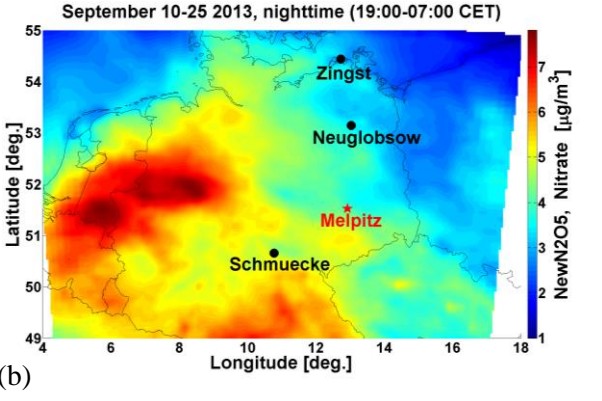

(b)

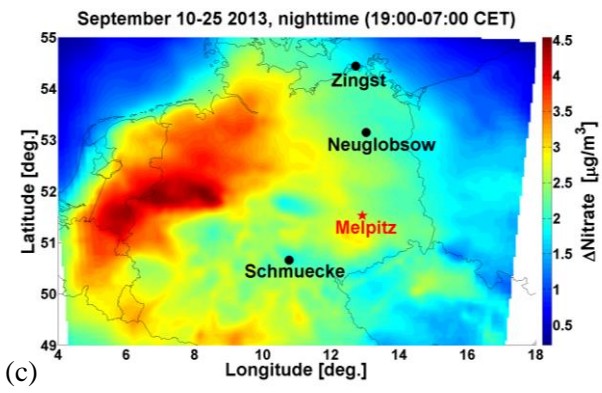

(c)

**Figure 5.** Horizontal distribution of averaged modelled particulate nitrate mass concentration during nighttime in September 10-25. (a) OldN2O5 case; (b) NewN2O5 case; (c) difference between OldN2O5 and NewN2O5 cases.

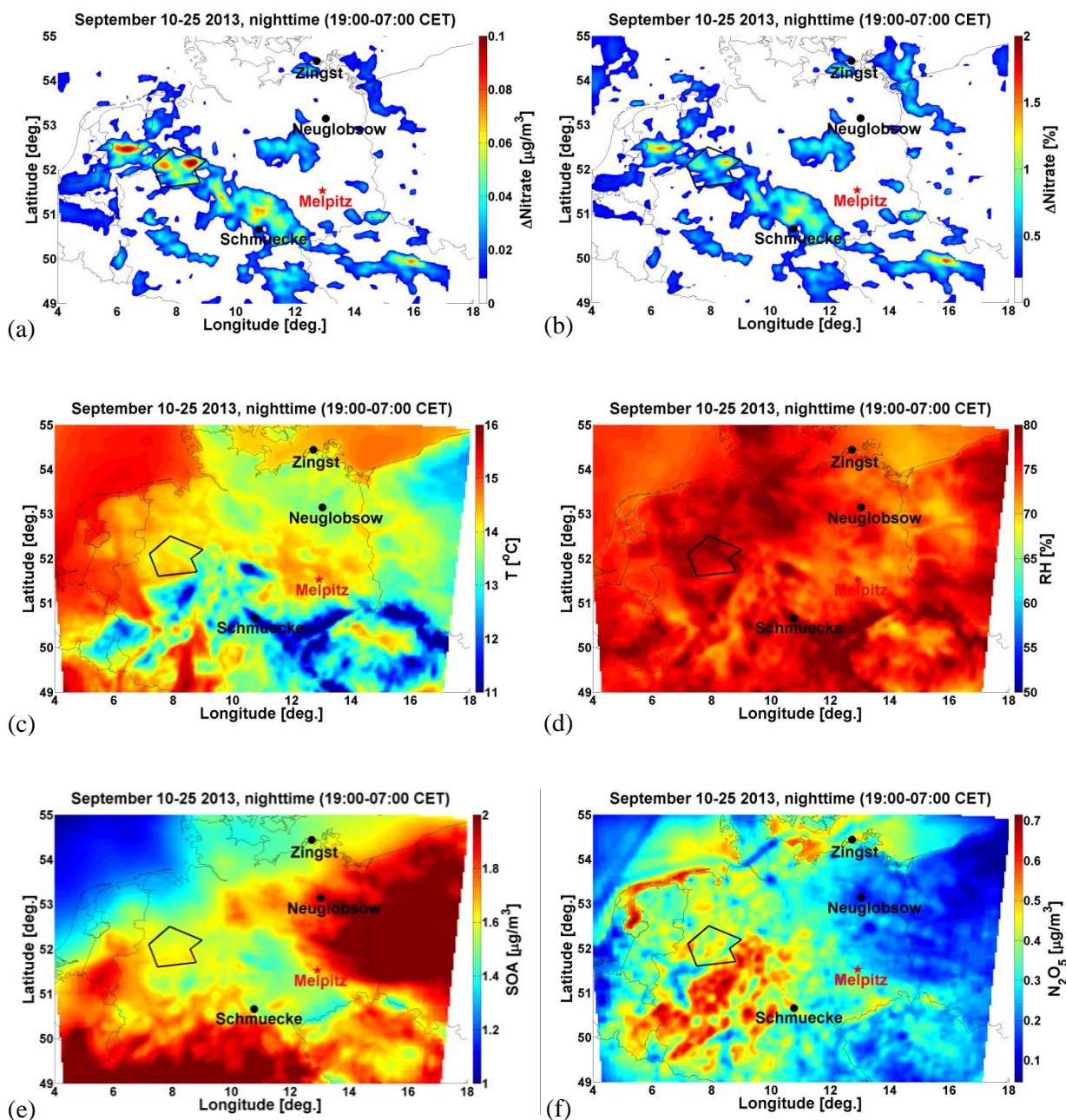

**Figure 6.** Horizontal distribution of averaged model results during nighttime in September 10-25, computed with the NewN2O5. (a) Difference of particulate nitrate mass concentration between model cases with and without considering organic coating effect; (b) difference of particulate nitrate mass concentration in percent between model cases with and without considering organic coating effect; (c) temperature; (d) RH; (e) SOA mass concentration; (f) $N_2O_5$ mass concentration. The region with the most significant organic coating effect is highlighted by the black polygon.