# Peer review of "A Parameterization of Heterogeneous Hydrolysis of N2O5 for Mass-based Aerosol Models: Improvement of Particulate Nitrate Prediction"

_Atmospheric Chemistry and Physics, 2017_

## Referee Comment (RC1) · Anonymous Referee #1 · 16 Apr 2017

Chen et al. have studied a new parameterization of heterogeneous hydrolysis of N2O5 within a 3D model over Germany. Clear improvement of using this parameterization with respect to original parameterizations is shown by comparing against measurements. Sensitivity tests have been performed to study the effect of NH3 emission, reaction constant and organic coating. The paper is well structured and easy for reading. It is recommended for publishing with minor revisions.

General Comments:

The measurement data used to evaluate the model performance are based on 24h filter sampler, but it is interesting to know the detailed temporal evolution at least in the model and have some discussion on the uncertainties related to NOx and N2O5 pre-

diction. This new parameterization of heterogeneous hydrolysis of N2O5 established from many previous laboratory experiments improves the prediction, but large gaps still exists between the model results and the measurement at all stations. Among the reasons given in section 3.1,how about the kN2O5 calculating with overestimated nitrate and what about its impact on the simulation?

Specific Comments:

1. P6, line 27, "is considered as 1.3 times of sulfate mass concentration", does this mean sulfate is not explicitly simulated in the model? What can be the "positive feedback" on line 29?

2. Table 1, SSA abbreviation is not introduced

3. P8, line 5: RH and wind speed have relatively important biais with respect to the measurement on 15-17 and 20-23 during the night. It should be discussed their relative impact on simulation results.

4. P8, line 20, Are the factors calculated based on average concentration during the campaign?

5. P8, line 22, is the 20-30% overestimation due to NH3 overestimation a conclusion from previous study?

6. P8, line 29, please quantify "significant"

7. P8, line 38, what does it mean "higher temporal resolution"

8. Figure 3, the shade cannot be clearly seen

9. Figure 4, why Melpitz is pointed in red?

10. P9, line 30, please quantify "more reasonable".

---

## Referee Comment (RC2) · Anonymous Referee #2 · 19 May 2017

This paper describes a new parametrization for the N2O5 hydrolysis which is dependent on T, RH and aerosol composition. The development of such a parameterization for N2O5 hydrolysis for a model framework that does not explicitly track aerosol surface area could be of interest to the community. However, the paper presented here has several serious shortcomings and errors, detailed below, so that I cannot recommend it for publication.

Major concerns:

1. The paper completely misrepresents the parameterization from Riemer et al. (2003): It is stated by the authors that the particle surface area in Riemer et al. (2003) was set to a constant value of 600 $\mu$m2 cm-3. However, this constant value was used only

for box model runs. In all other simulations (KAMM/DRAIS and EURAD) a constant value was not used, instead the mass, the number and thus the particle surface area were calculated with the modal aerosol model MADE and were highly variable. In Figures 8a and 11a of Riemer et al. (2003), examples of these variable horizontal distributions of the aerosol surface area density are depicted. It is clearly stated in Riemer et al. (2003) that even the corrected formula of Chang et al. (1987) shows a big difference in comparison to the more-complete parameterization that takes into account the dependence on aerosol surface area concentration (Sec. 4.2 in Riemer et al., 2003). The comparison with the so-called Riemer03 parametrization and an assumption of a reaction probability of 0.1 (Dentener and Crutzen, 1993) is not very helpful because numerous papers (e.g. Davis et al. (2008)) show that 0.1 is seen as an upper limit of gamma .

2. Chang et al. (1987) calculated the rate constant by the following equation: Eq. 17, Chang et al. (1987).

Whereas in this paper: Eq. 2 and 3, this study is written.

It is not clear whether this is an error in the paper, or also in the parameterization itself. It is not clear which formulation was the basis for the presented simulations. The authors need to check this because using the equation written in the paper gives values that are orders of magnitude different.

3. In equation 5, there is no explanation as to why the expression for gammaN2O5 is divided by a factor of 0.1. This leaves me with the impression that the factors are introduced to yield the best fit with the nitrate observations, which limits the general applicability of the parameterization to other domains and conditions. Similarly, there is a division by 600 in equation 4 which is also not explained. Furthermore, the units of fs are unclear. Based on the units stated in the text below equation 4, fs appears to have units of m-1 , but the factor should be unitless.

4. The reference to Chang et al. (2016) is missing. They also combined the Davis et

al. (2008) parametrization with the coating parameterization of Riemer et al. (2009).

Chang, W. L., S.S. Brown, J. Stutz, A.M. Middlebrook, R. Bahreini, N.L. Wagner, W.P. Dubé, I.B. Pollack, T. B. Ryerson, and N. Riemer (2016), Evaluating N2O5 heterogeneous hydrolysis parameterizations for CalNex 2010, J. Geophys. Res. Atmos., 121, 5051–5070,doi:10.1002/2015JD024737.
* * *

---

## Author Comment (AC1) · 28 Jun 2017

**Response to comments of referee #1**

**General Comments:**

Chen et al. have studied a new parameterization of heterogeneous hydrolysis of N2O5 within a 3D model over Germany. Clear improvement of using this parameterization with respect to original parameterizations is shown by comparing against measurements. Sensitivity tests have been performed to study the effect of NH3 emission, reaction constant and organic coating. The paper is well structured and easy for reading. It is recommended for publishing with minor revisions.

The measurement data used to evaluate the model performance are based on 24h filter sampler, but it is interesting to know the detailed temporal evolution at least in the model and have some discussion on the uncertainties related to NOx and N2O5 prediction. This new parameterization of heterogeneous hydrolysis of N2O5 established from many previous laboratory experiments improves the prediction, but large gaps still exists between the model results and the measurement at all stations. Among the reasons given in section 3.1, how about the kN2O5 calculating with overestimated nitrate and what about its impact on the simulation?

**Response:**

*Many thanks to the reviewer for the comments and suggestions.*

*This is a good suggestion. NOx is also an important precursor of nitrate. However, in this study, the overprediction of nitrate was not stem from NOx, which was in line with the measured concentration level. The detailed temporal evolutions of NOx and $N_2O_5$ were added in the supplementary information Text S1, as shown below.*

*"S1. Temporal evolutions of NOx and $N_2O_5$*

*The concentration of gaseous precursor (NOx) was observed under the frame of HOPE-Melpitz campaign with 1h temporal resolution. As shown in Fig. S3 (newly added), the modelled NOx concentration was in line with the measurement, with a factor of 0.9 for both OldN2O5 and NewN2O5 cases. Therefore, the high overestimation of particulate nitrate should not be resulted from the uncertainty of NOx.*

*The N$_2$O$_5$ concentration was accumulated during nighttime in NewN2O5 case, and was totally dissociated into NO$_2$ and NO$_3$ during daytime (Fig. S3b). However, the N$_2$O$_5$ could not accumulate during nighttime in OldN2O5 case, due to its highly overestimated reaction constant."*

[Figure]

**Figure S3 (newly added).** *Time series of NOx (a) and N$_2$O$_5$ (b) at Melpitz.*

*In addition, one sentence has been added in the manuscript (section 3.2) to summarize the above information, as shown below.*

*"The modelled NOx was in line with the observed concentration level at Melpitz, and should not be the reason of the overprediction of particulate nitrate (see details in Supplement Text S1 and Fig. S3)."*

*As reviewer mentioned, large gaps still exists between the model results and the measurements at all stations. However, this should not be stem from NewN2O5 scheme. Since, NewN2O5 may provide a $k_{N_2O_5}$ in the range of 0.36-1.2 times of the realistic one, as discussed in the newly added section 3.1. There must be some other reasons that are responsible for the remained large gaps. In addition to the reasons given in section 3.2 (revised version), the underprediction of coating organic matter budget in the model may also be a possible reason*

*(Chang et al., 2016). A sentence has been added in section 3.2 to include this information, as shown below.*

"One possible reason can be the underprediction of coating organic matter budget in the model leading to an overestimation of $\gamma_{N_2O_5}$ (Chang et al., 2016) " *has been* **added.**

*And the impact of overestimated nitrate was excluded when calculate kN2O5 in NewN2O5 scheme. In order to state this more clearly, we rephrased the description in section 2.2, as shown below.*

" $k_{\overline{N_2O_5}}$ ." **changed to**

*"Note that a small initial overestimation of particulate nitrate may result in a significant overprediction of nitrate, through the integration in models due to a feedback in this scheme. That is higher nitrate concentrations result in a larger $f_s$ and promise a higher $k_{N_2O_5}$, leads to a higher production of nitrate. In order to avoid the uncertainty of this feedback mechanism and to calculate a reasonable $k_{N_2O_5}$ in this case study, the nitrate mass concentration in equation (4) is considered as 1.3 times of sulfate mass concentration based on filter measurements during the HOPE-Melpitz campaign."*

**Specific Comments:**

(1) P6, line 27, "is considered as 1.3 times of sulfate mass concentration", does this mean sulfate is not explicitly simulated in the model? What can be the "positive feedback" on line 29?

**Response:**

*Thanks for the comment. The sulfate is simulated in the model. Here, we considered the nitrate as 1.3 times of sulfate mass concentration when calculate $k_{N_2O_5}$, is aiming to avoid the positive feedback mechanism in nitrate simulation, as described in General Comments. The corresponding statement has also been rephrased, as shown in General Comments.*

(2) Table 1, SSA abbreviation is not introduced.

**Response:**

*The introduction of sea salt aerosol (SSA) abbreviation has been added in Table 1.*

(3) P8, line 5: RH and wind speed have relatively important bias with respect to the measurement on 15-17 and 20-23 during the night. It should be discussed their relative impact on simulation results.

**Response:**

*Thanks for the comment. The discussion about the impact of RH and wind speed bias during the night on the particulate nitrate simulation has been added in the first paragraph of section 3, as shown below.*

*"Although model simulations slightly underestimated RH during the nighttime of September 17 and 22 (Fig. 2b), modelled RH was still higher than 80% where $k_{N_2O_5}$ is insensitive to RH as shown in Table 1 and Riemer et al. (2003). Therefore, this bias of RH will not lead to a significant uncertainty in nitrate simulation. However, the overestimation of wind speed may favour the transport of ammonia from Western Europe (e.g. the Netherlands). This could be a possible reason for the nitrate overprediction in NewN2O5 case (Fig. 3d), especially during September 20-24 when western wind was constantly dominant (Fig. 2d)."*

[Figure]

***Figure R1.*** *Rate constant for the heterogeneous hydrolysis of $N_2O_5$ with relation to RH. Modified from Figure 1 of Riemer et al. (2003), or calculated from the equation (2) with a=17.*

(4) P8, line 20, Are the factors calculated based on average concentration during the campaign?

**Response:**

*Yes, as reviewer understood, the factors are calculated based on the average concentration during the campaign.*

(5) P8, line 22, is the 20-30% overestimation due to NH3 overestimation a conclusion from previous study?

**Response:**

*Thanks for the comment. Yes, as reported in previous studies that a 50% ammonia emission reduction leads to a 16-50% reduction (Backes et al., 2016) or a maximum of 30% reduction (Renner and Wolker, 2010) of particulate nitrate concentration. These are in line with our result, and the corresponding sentence in section 3.1 has been modified to include this information. As shown below:*

*"" changed to*

*"This is in line with the previous studies (Renner and Wolke, 2010; Backes et al., 2016)."*

(6) P8, line 29, please quantify "significant".

**Response:**

*Thanks for the comment. The corresponding sentence has been modified, as shown below.*

*""changed to*

*"It is due to a significant decrease (by averagely more than a factor of 20, see Fig. 4) in the reaction constant of heterogeneous hydrolysis of $N_2O_5$ by NewN2O5"*

(7) P8, line 36, what does it mean "higher temporal resolution".

**Response:**

*Thanks for the comment. It means the filter measurements at Melpitz were operated every day, instead of the every third day at other UBA stations (Neuglobsow, Schmücke and Zingst). In*

*order to state this more clearly, the corresponding statement has been modified, as shown below.*

*"and the comparison with Melpitz measurements (Fig. 3d), which "* ***changed to***

*"and the comparison with Melpitz measurements (Fig. 3d), which were sampled on filter every day and off-line analyzed"*

(8) Figure 3, the shade cannot be clearly seen.

**Response:**

*Thanks for the comment. The shading colors have been deepened. As shown below.*

[Figure]

***Figure 3.*** *Comparison of particulate nitrate mass concentration between filter measurements and modelled results: (a) Neuglobsow; (b) Schmuecke; (c) Zingst; (d) Melpitz. Modelled concentrations at Melpitz: (e) $N_2O_5$; (f) marker species T1 for chemical reaction R1; (g) marker species for chemical formation of particulate nitrate (T3-T2); (h) the $NH_3$ marker tracer (T-NH3) for transport from the Netherlands and south Germany. The light-red colour bars indicate the results of OldN2O5-FullNH3 case; the red colour bars indicate the results of OldN2O5 case; and the blue colour bars indicate the results of NewN2O5 case. The shaded periods indicate the dominating processes for high concentrations of particulate nitrate: chemical formation (red), transport (blue), and boundary conditions (grey).*

(9) Figure 4, why Melpitz is pointed in red?

**Response:**

*Thanks for the comment. Melpitz is pointed in red in Figure 1, Figure 4 and Figure 5, since its results were detailed discussed in section 3.1, Figure 2 and Figure 3. This information has been added in the caption of Figure 1, as shown below.*

*"Neuglobsow, Schmücke and Zingst are marked by black dots; Melpitz is marked in a red star and its results will be detailed discussed in Fig. 2 and Fig. 3."*

(10) P9, line 31, please quantify "more reasonable".

**Response:**

*Thanks for the comment. The corresponding sentence has been modified, as shown below.*

*"Therefore, the regions with high $[NO_3^-]$ during nighttime indicates considerable nitrate formation from the heterogeneous hydrolysis of $N_2O_5$, which was reduced to a more reasonable value in our new scheme." **changed to***

*"Therefore, the regions with high $[NO_3^-]$ during nighttime indicates a considerable nitrate formation from the heterogeneous hydrolysis of $N_2O_5$, where $[NO_3^-]$ was reduced by about 3-4.5 $\mu g/m^3$ (~35%, see Fig. 5) in the new scheme."*

**Reference:**

Backes, A. M., Aulinger, A., Bieser, J., Matthias, V., and Quante, M.: Ammonia emissions in Europe, part II: How ammonia emission abatement strategies affect secondary aerosols, Atmospheric Environment, 126, 153-161, http://dx.doi.org/10.1016/j.atmosenv.2015.11.039, 2016.

Chang, W. L., Brown, S. S., Stutz, J., Middlebrook, A. M., Bahreini, R., Wagner, N. L., Dubé, W. P., Pollack, I. B., Ryerson, T. B., and Riemer, N.: Evaluating N2O5 heterogeneous hydrolysis parameterizations for CalNex 2010, Journal of Geophysical Research: Atmospheres, 121, 5051-5070, 10.1002/2015JD024737, 2016.

Davis, J. M., Bhave, P. V., and Foley, K. M.: Parameterization of N2O5 reaction probabilities on the surface of particles containing ammonium, sulfate, and nitrate, Atmos. Chem. Phys., 8, 5295-5311, 10.5194/acp-8-5295-2008, 2008.

Renner, E., and Wolke, R.: Modelling the formation and atmospheric transport of secondary inorganic aerosols with special attention to regions with high ammonia emissions, Atmospheric Environment, 44, 1904-1912, http://dx.doi.org/10.1016/j.atmosenv.2010.02.018, 2010.

Riemer, N., Vogel, H., Vogel, B., Schell, B., Ackermann, I., Kessler, C., and Hass, H.: Impact of the heterogeneous hydrolysis of N2O5 on chemistry and nitrate aerosol formation in the lower troposphere under photosmog conditions, Journal of Geophysical Research: Atmospheres, 108, n/a-n/a, 10.1029/2002JD002436, 2003.

---

## Author Comment (AC2) · 28 Jun 2017

**Response to comments of referee #2**

**General Comments:**

This paper describes a new parametrization for the $N_2O_5$ hydrolysis which is dependent on T, RH and aerosol composition. The development of such a parameterization for $N_2O_5$ hydrolysis for a model framework that does not explicitly track aerosol surface area could be of interest to the community. However, the paper presented here has several serious shortcomings and errors, detailed below, so that I cannot recommend it for publication.

**Response:**

*Thanks to the reviewer for the comments and suggestions. Yes, as emphasized by the reviewer, this work proposed a parameterization for $N_2O_5$ hydrolysis for the computationally efficient mass-based aerosol models. This work can be very useful for some regional models (e.g. COSMO-MUSCAT with aerosol treatment based on Simpson et al., 2003), and also for some global models (e.g. HadGEM3-ES, Bellouin et al. 2011; Hardiman et al., 2017). In order to emphasize this, we modified the title, as shown below.*

*"A Parameterization of Heterogeneous Hydrolysis of $N_2O_5$ for 3-D Atmospheric Modelling: Improvement of Particulate Nitrate Prediction"* **changed to**

*"A Parameterization of Heterogeneous Hydrolysis of $N_2O_5$ for Mass-based Aerosol Models: Improvement of Particulate Nitrate Prediction"*

*The manuscript has been modified accordingly. Please find the detailed point-to-point modifications and corrections in the following.*

**Major concerns:**

(1.1) The paper completely misrepresents the parameterization from Riemer et al. (2003): It is stated by the authors that the particle surface area in Riemer et al. (2003) was set to a constant value of 600 μm2 cm-3. However, this constant value was used only for box model runs. In all other simulations (KAMM/DRAIS and EURAD) a constant value was not used, instead the mass, the number and thus the particle surface area were calculated with the modal aerosol

model MADE and were highly variable. In Figures 8a and 11a of Riemer et al. (2003), examples of these variable horizontal distributions of the aerosol surface area density are depicted. It is clearly stated in Riemer et al. (2003) that even the corrected formula of Chang et al. (1987) shows a big difference in comparison to the more-complete parameterization that takes into account the dependence on aerosol surface area concentration (Sec. 4.2 in Riemer et al., 2003).

**Response:**

*We apologize for the misleading introduction of the parameterization from Riemer et al. (2003), who proposed two parameterizations (P1 and P2) as shown in Table R1 and Fig. R1. The reviewer was right, the particle surface area (S) was comprehensively considered in P1, but was not considered in P2. P2 was only used in the box model (Figure 2 of Riemer et al., 2003) and 1-D simulations (Figure 5 of Riemer et al., 2003). Riemer et al. (2003) suggested to use 'a=17' instead of 'a=5' (suggested by Chang et al., 1987) for a better approximation of the more realistic P1. This produces a result that is very close to the P1 with 'S=600 $\mu m^2$ $cm^{-3}$' when RH>60%, as shown in Fig. R1. Here, we only adopted and improved the P2. However, we mistakenly named P2 as 'Riemer03' with a constant 'S=600 $\mu m^2$ $cm^{-3}$', which is inappropriate for introducing parameterizations from Riemer et al. (2003). We changed the 'Riemer03 scheme' to 'Original scheme of COSMO-MUSCAT' or 'OldN2O5', and corrected the corresponding context throughout the manuscript, as shown later.*

*The sophisticated P1 is more suitable for models with complex aerosol treatment, e.g. KAMM/DRAIS and EURAD with the modal aerosol module (Riemer et al. 2003) or WRF-Chem with a sectional aerosol module MOSAIC (Chang et al., 2016). However, the simulation of particle surface area is still a challenging task even in the models with complex aerosol treatment. As mentioned in Chang et al. (2016), the aerosol liquid water need to be considered when estimating its particle surface area. However, the aerosol thermodynamic models may not accurately capture aerosol liquid water at low RH.*

*Nevertheless, P2 is very suitable for the computationally efficient mass-based aerosol models (as described in the reply of General Comments), where P1 is difficult to be adopted. Therefore, P2 is used in the current version of COSMO-MUSCAT. Unfortunately, as pointed out by the reviewer, even the corrected formula of Chang et al. (1987) (P2 with 'a=17') shows a big difference in comparison to the more-complete parameterization (P1) that takes into account the dependence on aerosol surface area concentration. This shows the*

*importance of our work: to propose/improve a parametrization for the $N_2O_5$ hydrolysis that is suitable for the computationally efficient mass-based aerosol modules, without a big compromise of accuracy.*

**Table R1.** *Parameterizations (P1 and P2) from Riemer et al. (2003).*

$$k_{N_2O_5} = \frac{1}{4} \cdot c_{N_2O_5} \cdot S \cdot \gamma_{N_2O_5}, \qquad (\text{P1})$$

*Where $kN_2O_5$ is the reaction constant, $cN_2O_5$ is the mean molecular velocity of $N_2O_5$, $\gamma N_2O_5$ is the reaction probability, and S is the aerosol surface area density*

$$k_{N_2O_5} = \frac{1}{600 \ \exp\left(-\left(\frac{RH}{28}\right)^{2.8}\right) + a}. \qquad (\text{P2})$$

*where RH is the relative humidity in % and $k_{N_2O_5}$ results in min$^{-1}$.*

[Figure]

**Figure R1.** *Rate constants for the heterogeneous hydrolysis of $N_2O_5$ when different parameterizations are used. Source: Figure 1 of Riemer et al. (2003)*

*We changed the description of parameterizations in Riemer et al. (2003), as shown below.*

***In the 'Introduction' part:***

[revised manuscript text omitted]

*"*

* However, the complex considerations of S and $\gamma_{N_2O_5}$ is still missing in the OldN2O5.*

*In this study, we proposed a sophisticated parameterization to improve the OldN2O5 for mass-based aerosol models, with  full consideration of temperature, RH, aerosol particle compositions and S."*

*The 'Riemer03' in the 'Results & Discussion' and 'Conclusion' parts of the original manuscript were revised to 'OldN2O5' accordingly. Please find the detailed corrections in the revised manuscript with track changes.*

(1.2) The comparison with the so-called Riemer03 parametrization and an assumption of a reaction probability of 0.1 (Dentener and Crutzen, 1993) is not very helpful because numerous papers (e.g. Davis et al. (2008)) show that 0.1 is seen as an upper limit of gamma.

**Response:**

*Thanks for the comment. In this work, we compared NewN2O5 with the P2 ('a=17', Riemer et al., 2003) which is currently used in COSMO-MUSCAT. In the original manuscript, we called*

*it a comparison with an assumption of 'γN2O5=0.1', since P2 is developed on basis of 'γN2O5 =0.1', as described in Riemer et al (2003):*

*"The parameterization P2 is based on the assumption that the relative humidity is an indicator for the aerosol surface area density and that γN2O5= 0.1." from page 5-3 of Riemer et al (2003).*

*However, we agree with the reviewer that this interpretation is misleading and confusing. Therefore, we renamed this comparison to 'the comparison with the original parameterization of COSMO-MUSCAT', and modified corresponding texts throughout the manuscript, as shown later. And the Figure S2 (in the original manuscript) is replaced by a more interesting comparison with Chang et al. (2016). This was added in the section 3.1 of the revised manuscript, as described below.*

*Chang et al. (2016) also used Davis et al. (2008) and Anttila et al. (2006) to estimate the γN2O5. Their study adopted P1 (Riemer et al., 2003) into WRF-Chem with a sectional aerosol treatment (MOSAIC). Their results were validated by the aircraft measurements ($γ_{ss}$, estimated reaction probability in steady state) in the CalNex-2010 campaign, and showed a reasonable result (Fig. 4). In order to validate the performance of our mass-based parameterization (NewN2O5), we performed the simulation with WRF-Chem (MOSAIC) during the HOPE-Melpitz campaign. The WRF-Chem results with 8 aerosol size bins (40 nm to 10 μm) were carried out for off-line estimations (see a new method section 2.3) of k_sectional (reaction constant according to Chang et al. 2016, y-axis in Fig. S2) and mass-based k_NewN2O5 (according to our parameterization, x-axis in Fig. S2). The comparison between k_sectional and k_NewN2O5 shows a good agreement (R=0.91), although k_NewN2O5 may be lower by a factor of ~8 (Fig. S2) than k_sectional. The possible reasons for this difference and the uncertainties are discussed in the section 3.1 of the revised manuscript, as shown later. This comparison further approves a reasonable performance of our mass-based NewN2O5 parameterization.*

*Although Chang et al. (2016) reported that 'Davis+coat' provided the best results compared with observations, here, we validated our NewN2O5 with the 'Davis' (without OC coating) according to Chang et al. (2016), namely Ch&Davis in the revised manuscript, due to the following reasons:*

*(1) HOPE-Melpitz campaign is an OC-low (less than 7%) case, there is hence not so much organic carbon (OC) available for coating. Therefore, validating with a non-coating parameterization (Ch&Davis) would be more reasonable. Furthermore, the OC coating effect will only make a difference less than 1% at Melpitz during our case.*

*(2) The treatments of OC coating are different between Chang et al. (2016) and NewN2O5, although we both used Anttila et al. (2006) scheme. In Chang et al. (2016), total OC (Primary OC + SOA) was treated as OC coating. However, our NewN2O5 only treat SOA for coating, which should be more reasonable and is consistent with the original literature Riemer et al. (2009).*

*(3) Chang et al. (2016) used the WRF-Chem (V3.3.1) with CBMZ-MOSAIC scheme, which does not consider the formation of SOA, as described by the MOSAIC developer (Zaveri et al., 2008). However, in this study, we would like to adopt NewN2O5 scheme into COSMO-MUSCAT, which treat SOA formation based on SORGAM (Schell et al., 2001; Li et al., 2013).*

*(4) The equation (11) written in the Chang et al. (2016) is not identical with its citation (Riemer et al., 2009), which described the OC coating effect according to Anttila et al. (2006). As shown following:*

$$\gamma_{coat} = \frac{4RTH_{org}D_{org}R_c}{c_{N_2O_5}} \cdot \ell \cdot R_p$$

*,   (Eq. 11 in Chang et al. 2016)*

$$\gamma_{i,coat} = \frac{4RTH_{org}D_{org}R_{c,i}}{c_{N_2O_5}\ell_i R_{p,i}}$$

*,   (Eq. 6 in Riemer et al. 2009, also Eq. 11 in Anttila et al. 2006)*

*We believe that it is just a typo in the paper, and the model simulations were correctly calculated in Chang et al. (2016). However, to make sure that our validation is completely reliable and to avoid unnecessary confusion, we would prefer to validate our results with the 'Davis' (without OC coating) according to Chang et al. (2016), namely Ch&Davis in the revised manuscript.*

*A new section 3.1 was added in the revised manuscript to validate the mass-based NewN2O5 parameterization with the more sophisticated sectional-based approach according to Chang et al., (2016), as shown below.*

*"**3.1 Evaluating closure for mass-based NewN2O5 and a sectional approach**

In order to confirm that the mass-based NewN2O5 estimates $k_{N_2O_5}$ with a reliable accuracy, we evaluated closure between NewN2O5 and a sectional-based state-of-the-art parameterization (Chang et al., 2016) based on the WRF-Chem (MOSAIC) results. Chang et al. (2016) reported that 'Davis + coat' (Daivs08 + Anttila06) approach produced a best agreement of $\gamma_{N_2O_5}$ with aircraft observations during the CalNex-2010 campaign, with overestimation by a factor mostly within in a range of 2-8 (Fig. S2b). Without considering OC coating effect (Davis08 only), the Ch&Davis still showed a relatively good linear relationship with the observed $\gamma_{N_2O_5}$, which was however overestimated with a higher factor ranging about 3-10 (Fig. S2a). Considering the different treatments of OC coating between NewN2O5 (SOA coating only) and Chang et al. (2016) ('Davis + coat', Primary OC and SOA), the NewN2O5 was validated using the Ch&Davis scheme. This would not significantly influence the comparison results, since the HOPE-Melpitz campaign was an OC-low case, with only ~7% contribution from total OC mass based on filter measurements at Melpitz. Therefore, not much SOA was available for coating effect, different to the OC-high case (contributed about 50-80% to total mass, Figure 9 in Chang et al., 2016) in the CalNex-2010 campaign. The coating effect exerted a negligible influence at Melpitz, this point will be discussed in detail in section 3.4. We validated NewN2O5 scheme by comparing $k_{N_2O_5}$ instead of $\gamma_{N_2O_5}$, because NewN2O5 scheme was developed on basis of a parameterization to directly calculate $k_{N_2O_5}$ proposed by Chang et al. (1987) and Riemer et al (2003).

As shown in Fig. 4, the $k_{N_2O_5}$ showed a very good linear relationship (R=0.91) between NewN2O5 and Ch&Davis, much better than using the OldN2O5 (Fig. 4). Mass-based NewN2O5 estimated lower $k_{N_2O_5}$ than the sectional-based Ch&Davis by a factor of ~8. However, Ch&Davis may overestimate the $\gamma_{N_2O_5}$ by a factor of 3-10 (Chang et al, 2016, see also Fig. S2a). Assuming that S was correctly given by the WRF-Chem sectional aerosol module, we can expect that Ch&Davis may overestimate $k_{N_2O_5}$ by a factor of 3-10 according to the equation (1). Therefore, NewN2O5 may provide a $k_{N_2O_5}$ in the range of 0.36-1.2 times of the realistic one.*

*Two important uncertainties are needed to be kept in mind in this validation. First, the estimation of S is very challenging, due to the uncertainties of particle number/mass size distribution, partitioning processes, secondary formation and etc. In addition, the hygroscopic grow of particle can also be an important source of the uncertainty of S, due to the challenge in the estimation of particle liquid water especially at low RH, even by a complex aerosol treatment (Chang et al., 2016). About 30% difference of $k_{N_2O_5}$ between NewN2O5 and Ch&Davis is stem from the different treatments of S. As shown in Fig. 4, the factor between NewN2O5 and Ch&Davis reduced from ~8.3 to ~5.9, with a slightly increase of R, when we adopted the sectional-based S (same as Ch&Davis) in NewN2O5. Second, the Ch&Davis was validated by aircraft measurements in an OC-high case during the CalNex-2010 campaign. Therefore, the overestimation factor of Ch&Davis may not be as high as expected in an OC-low case during the HOPE-Melpitz campaign. However, the SSA, BC and dust should exert a sensible influence in an OC-low case, and should also be considered in a parameterization, as we did in NewN2O5. This can be also a reason for the difference between Ch&Davis and NewN2O5."*

[Figure]

***Figure 4 (newly added).*** *Comparison between the sectional-based Ch&Davis ('Davis' of Chang et al., 2016) and mass-based NewN2O5 (mass-based, black), NewN2O5 (with a sectional-based particle surface area, blue) and OldN2O5 (red). The results are offline calculated on basis of WRF-Chem simulation with a sectional aerosol treatment (MOSAIC).*

[Figure]

**(a)**                                      **(b)**

*Figure S2 (**newly added**). Modelled γ (Chang et al., 2016) versus calculated γ$_{ss}$ (reaction probability in steady state) using aircraft observations from the 31 May flight of CalNex 2010 campaign. (a) Davis (Davis et al., 2008, namely Ch&Davis) and B&T (Bertram and Thornton, 2009) parameterization; (b) Davis+coat (Anttila06).*

*Source: Figure 10 of Chang et al. (2016).*

*A new section 2.3 was added in the revised version (as shown below), to describe the WRF-Chem simulations that we used for evaluating closure between NewN2O5 and Chang et al. (2016). The incorporation of mass-based NewN2O5 to the sectional aerosol module in WRF-Chem is also described.*

*"**2.3 Estimation of reaction probabilities with a sectional aerosol model***
*The Weather Research and Forecasting/Chemistry model (WRF-Chem V3.5.1) is a fully on-line coupled regional air quality model. Chang et al. (2016) incorporated several parameterizations for the $N_2O_5$ hydrolysis into a sectional aerosol treatment (MOSAIC, Zaveri et al., 2008) in WRF-Chem. 'Davis' approach from Chang et al. (2016), hereinafter referred to as Ch&Davis, was chosen to be compared with NewN2O5. The reasons for this choice will be discussed in detail in section 3.1.*

*In order to validate the mass-based NewN2O5 with the sectional-based Ch&Davis, we performed WRF-Chem simulation during the HOPE-Melpitz campaign. The same WRF-Chem results were adopted for offline estimating $k_{N_2O_5}$ according to NewN2O5 and Ch&Davis,*

*respectively. We followed the physics relating configuration according to Chen et al. (2016a), which well reproduced meteorological conditions during the HOPE-Melpitz campaign. The sea salt emission (Gong, 2003) was reduced by a factor of 20 in WRF-Chem, considering that Gong (2003) may highly overestimate sea salt emission (Neumann et al., 2016), and thus leads to an overestimation of sea salt by a factor of 20 during the HOPE campaign at Melpitz (Chen et al., 2016b). The configuration of chemical and aerosol treatments followed Chang et al. (2016). CBMZ (Zaveri and Peter, 1999) mechanism was used to describe gas-phase reactions. MOSAIC (Zaveri et al., 2008) with eight size bins was chosen to represent aerosol properties. Three nested domains (Fig. S1) with 39 vertical layers were set up for the simulated case, with a resolution of 54 km, 18 km and 6 km respectively.*

*In Ch&Davis the aerosol liquid water is considered when calculating particle surface area for each size bin. Details of the sectional-based method for estimating S in Ch&Davis scheme are given by Chang et al. (2016). In NewN2O5 scheme, the first six bins (with diameter in the range of 40nm – 2.5 μm) are counted as fine mode, and the last two bins (2.5 -10 μm) are counted as coarse mode. This definition is identical with COSMO-MUSCAT. In order to be consistent with COSMO-MUSCAT, the organic coating effect is considered for fine particles in NewN2O5, since the maximum effective particle diameter of Anttila06 scheme is 2 μm (Anttila et al., 2006). In order to quantify the uncertainty stem from the different S treatments between NewN2O5 (mass-based) and Ch&Davis (sectional-based), an estimation result according to an adapted NewN2O5 (with sectional-based S) will also be discussed in section 3.1."*

[Figure]

***Figure S1 (newly added).*** *Domain setting of WRF-Chem simulation.*

(2) Chang et al. (1987) calculated the rate constant by the following equation: Eq. 17, Chang et al. (1987). Whereas in this paper: Eq. 2 and 3, this study is written. It is not clear whether this is an error in the paper, or also in the parameterization itself. It is not clear which formulation was the basis for the presented simulations. The authors need to check this because using the equation written in the paper gives values that are orders of magnitude different.

**Response:**

*Thanks a lot for pointing out the typo in Eq. 2 and 3. We have double checked that in the model, the equation is identical with the Eq. 17 in Chang et al. (1987). The calculated kN2O5 is given in Fig. R2, which is identical with the Figure 1 in Riemer et al. (2003) (see also Fig. R1). We apologize for the mistake, and corrected the equations, as shown below.*

[Figure]

**Figure R2.** *Rate constant for the heterogeneous hydrolysis of $N_2O_5$ with relation to RH. Modified from Figure 1 of Riemer et al. (2003), or calculated from the equation (2) with a=17.*

$$k_{N_2O_5} = \frac{1}{600\exp(-(\frac{RH}{28})^{2.8} + a)}$$ (2)

*changed to*

$$k_{N_2O_5} = \frac{1}{600\exp(-(\frac{RH}{28})^{2.8}) + a}$$ (2)

$$k_{N_2O_5} = \frac{1}{600\exp(-(\frac{RH}{28})^{2.8}+17)} \cdot f_s \cdot f_{\gamma_{N_2O_5}}$$

**changed to**

$$k_{N_2O_5} = \frac{1}{600\exp(-(\frac{RH}{28})^{2.8})+17} \cdot f_s \cdot f_{\gamma_{N_2O_5}}$$ (3)

(3) In equation 5, there is no explanation as to why the expression for gammaN2O5 is divided by a factor of 0.1. This leaves me with the impression that the factors are introduced to yield the best fit with the nitrate observations, which limits the general applicability of the parameterization to other domains and conditions. Similarly, there is a division by 600 in equation 4 which is also not explained. Furthermore, the units of fs are unclear. Based on the units stated in the text below equation 4, fs appears to have units of m$^{-1}$ , but the factor should be unitless.

**Response:**

*Thanks for the comment. In this study, we would like to propose a mass-based parameterization (NewN2O5) based on the P2 (Riemer et al., 2003). This NewN2O5 is the best approximation of P1 (Riemer et al., 2003), which is with respect to reaction probability (γ) and particle surface area concentration (S). Therefore, we introduced two factors ( $f_s$ and*

*$f_{\gamma_{N_2O_5}}$ ) to adjust the kN2O5 according to P2. The $f_{\gamma_{N_2O_5}}$ is calculated as Eq. 5, which is used to adjust the impact of γ. The $f_s$ is calculated as Eq. 4, which is used to adjust the impact of particle surface area (S).*

*As described in Riemer et al. (2003), P2 is developed on basis of the assumption 'γN2O5 = 0.1', and 'a=17' will provide a result that is very close to the more-complete P1 with 'S~= 600 μm$^2$ cm$^{-3}$' when RH is higher than 60% (see Fig. R1). Therefore, when we calculate the correction factors in NewN2O5, we divide γN2O5 by 0.1 and divide particle surface area (S) by 600 μm$^2$ cm$^{-3}$.*

*The original text is given: "The parameterization P2 is based on the assumption that the relative humidity is an indicator for the aerosol surface area density and that γN2O5 = 0.1. In*

*addition, the values of kN2O5 as they follow from P1 for different aerosol surface area densities (S = 200 μm² cm⁻³ and S = 600 μm² cm⁻³) are given in Figure 1. Although the aerosol surface area density is far from being constant in the real atmosphere, we included the curves based on P1 for comparison. P1 will be identical to P2 at high relative humidity (RH > 60%), if the surface area density is about 2700 μm² cm⁻³. However, such surface area densities can only be expected in highly polluted areas or if cloud droplets are present. Therefore P2 overestimates kN2O5 under cloud free and unpolluted conditions. If we use a = 17 instead of a = 5 in P2, it is a much better approximation for P1, as can be seen from Figure 1 (see also Fig. R1 of this response)" page 5-3 from Riemer et al. (2003).*

*However, we agree with the reviewer that the descriptions of Eq. 4 and Eq. 5 are not clear enough. We modified the descriptions and equations, and the $f_s$ is unitless, as shown below.*

$$f_{\gamma_{N_2O_5}} = (\gamma_{core}^{-1} + \gamma_{coating}^{-1})^{-1} / 0.1 \qquad (5)$$

**changed to:**

$$f_{\gamma_{N_2O_5}} = (\gamma_{core}^{-1} + \gamma_{coating}^{-1})^{-1} / \gamma_{ref} \qquad (5)$$

*where $\gamma_{core}$ is the N₂O₅ reaction probability with the core of the particle, which can be estimated by Table 1; $\gamma_{coating}$ is the N₂O₅ reaction probability with the secondary organic coating shell of the particle, which can be estimated by the formula (6) according to Anttila et al. (2006) and Riemer et al. (2009); $\gamma_{ref}$ is the reference reaction probability. Here, we suggest '$\gamma_{ref} = 0.1$', since Eq. 2 is developed on basis of the assumption '$\gamma_{N_2O_5} = 0.1$' (Riemer et al., 2003).*

$$f_s = (SA_{fine} \cdot PM_{fine} + SA_{coarse} \cdot PM_{coarse}) / 600 \qquad (4)$$

*where $SA_{fine}$ / $SA_{coarse}$ is the specific surface area for fine/coarse mode particles in $m^2/g$, $PM_{fine}$ / $PM_{coarse}$ is the mass concentration of fine/coarse mode particles in $\mu g/m^3$. A value 11 $m^2/g$ was used for $SA_{fine}$, considering recently reported values of 11.9 $m^2/g$ and 10.2 $m^2/g$ from laboratory studies (Okuda, 2013) and measurements in Japanese urban regions (Hatoya et al., 2016). A value of 1.2 $m^2/g$ was used for $SA_{coarse}$ (Okuda, 2013).*

**changed to:**

$$f_s = (SA_{fine} \cdot PM_{fine} + SA_{coarse} \cdot PM_{coarse}) / S_{ref} \qquad (4)$$

*where $SA_{fine}$ / $SA_{coarse}$ is the specific surface area for fine/coarse mode particles in $m^2/g$, $PM_{fine}$ / $PM_{coarse}$ is the mass concentration of fine/coarse mode particles in $\mu g/m^3$. A value 11 $m^2/g$ was used for $SA_{fine}$, considering recently reported values of 11.9 $m^2/g$ and 10.2 $m^2/g$ from laboratory studies (Okuda, 2013) and measurements in Japanese urban regions (Hatoya et al., 2016). A value of 1.2 $m^2/g$ was used for $SA_{coarse}$ (Okuda, 2013). $S_{ref}$ is the reference particle surface area concentration, here, we suggest '$S_{ref} = 600\ \mu m^2\ cm^{-3}$'. Since Eq. 2 will provide a result that is very close to a complex parameterization with 600 $\mu m^2\ cm^{-3}$ particle surface area concentration (Riemer et al., 2003), when 'a=17' and 'RH>60%'.*

(4) The reference to Chang et al. (2016) is missing. They also combined the Davis et al. (2008) parametrization with the coating parameterization of Riemer et al. (2009). Chang, W. L., S.S. Brown, J. Stutz, A.M. Middlebrook, R. Bahreini, N.L. Wagner, W.P. Dubé, I.B. Pollack, T. B. Ryerson, and N. Riemer (2016), Evaluating $N_2O_5$ heterogeneous hydrolysis parameterizations for CalNex 2010, J. Geophys. Res. Atmos., 121, 5051–5070,doi:10.1002/2015JD024737.

**Response:**

*Thanks for the very helpful latest study. This literature (Chang et al., 2016) has been added in the references. The differences of focuses, different applications and the comparison between our NewN2O5 and Chang et al. (2016) have been discussed in detail, as shown above.*

**Reference:**

[revised manuscript text omitted]